# Text Alignment Is An Efficient Unified Model for Massive NLP Tasks

**Yuheng Zha**[*]  **Yichi Yang**[*]  **Ruichen Li**  **Zhiting Hu**
UC San Diego
{yzha, yiy067, rul014, zhh019}@ucsd.edu

## Abstract

Large language models (LLMs), typically designed as a function of next-word prediction, have excelled across extensive NLP tasks. Despite the generality, next-word prediction is often not an *efficient* formulation for many of the tasks, demanding an extreme scale of model parameters (10s or 100s of billions) and sometimes yielding suboptimal performance. In practice, it is often desirable to build more efficient models—despite being less versatile, they still apply to a substantial subset of problems, delivering on par or even superior performance with much smaller model sizes. In this paper, we propose *text alignment* as an efficient unified model for a wide range of crucial tasks involving text entailment, similarity, question answering (and answerability), factual consistency, and so forth. Given a pair of texts, the model measures the degree of alignment between their information. We instantiate an alignment model (ALIGN) through lightweight finetuning of RoBERTa (355M parameters) using 5.9M examples from 28 datasets. Despite its compact size, extensive experiments show the model's efficiency and strong performance: **(1)** On over 20 datasets of aforementioned diverse tasks, the model matches or surpasses FLAN-T5 models that have around 2x or 10x more parameters; the single unified model also outperforms task-specific models finetuned on individual datasets; **(2)** When applied to evaluate factual consistency of language generation on 23 datasets, our model improves over various baselines, including the much larger GPT-3.5 (ChatGPT) and sometimes even GPT-4; **(3)** The lightweight model can also serve as an add-on component for LLMs such as GPT-3.5 in question answering tasks, improving the average exact match (EM) score by 17.94 and F1 score by 15.05 through identifying unanswerable questions.[2]

## 1  Introduction

Recent large language models (LLMs) have demonstrated exceptional generalizability in a wide range of natural language processing (NLP) tasks. As the underlying formulation of these LLMs, *next-word prediction* is proven to be a general function applicable to diverse language problems. However, it is often not being an *efficient* solution for many tasks. LLMs often need to scale up to over tens of billions of parameters to achieve meaningful performance [1], with popular models like GPT-3 boasting as many as 175B parameters [2]. Additionally, even with their extreme scale, LLMs sometimes still find themselves outperformed by smaller models. For example, ChatGPT/GPT-3.5 falls behind existing finetuned baselines on most classical natural language understanding tasks [3].

As a result, in many cases it is desirable to navigate the spectrum of generality-vs-efficiency tradeoff, for example, by developing smaller but general-purpose models that excel in a *substantial subset* of tasks. Despite being less versatile than the extreme-scale LLMs, these models are more efficient and

---

[*]Equal contribution.
[2]Code is made available at https://github.com/yuh-zha/Align

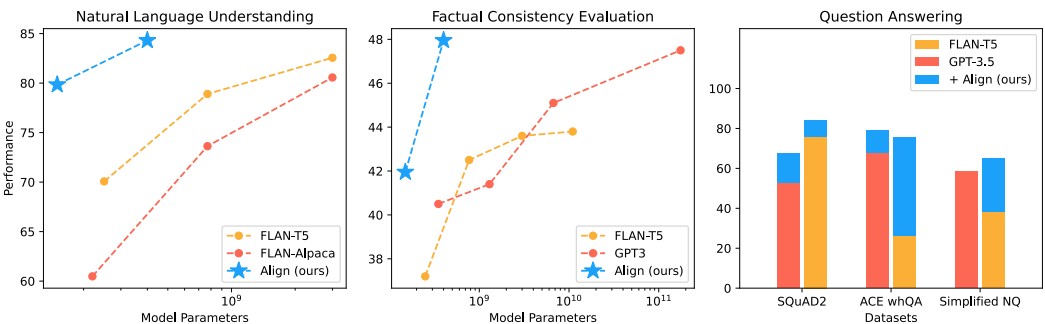

Figure 1: Our alignment model (125M and 355M) achieves substantially better efficiency and performance compared to much larger models on a wide range of tasks, including **(left)** diverse text pair understanding tasks on over 20 datasets and **(middle)** factual consistency evaluation, and **(right)** improves existing LLMs on question answering by detecting unanswerable questions. See Section 4 for more details.

provide superior performance, making them more usable on the set of tasks that they are designed to handle. Previous work has attempted to build natural language inference (NLI) models as an efficient solution for broad tasks [4–6]. But with limited NLI data (e.g., MNLI [7]) for training, the models exhibit limited performance and applicability across diverse domains. Another related line of research trains general text representation models with pretraining and multi-task learning [8–10]. Those models need to be specifically finetuned (with task-specific head) for each downstream task, instead of functioning as ready-to-use solutions.

In this paper, we investigate the underlying commonalities among a broad range of NLP tasks that concern the relationship between two texts, and propose a *text alignment* model (ALIGN) as an efficient unified solution, following Zha et al. [11]. Given an arbitrary pair of texts, ALIGN measures the degree of alignment between the content in the texts. We show the formulation subsumes a substantial set of popular tasks, ranging from NLI, fact verification, semantic textual similarity, question answering, coreference resolution, paraphrase detection, to factual consistency evaluation of diverse language generation tasks, question answerability verification, and so forth (Figure 1). The generality, in turn, presents an opportunity for us to use diverse data to learn the alignment model. Specifically, we adapt and aggregate 28 datasets from the above tasks, resulting in 5.9M training samples with diverse characteristics. We then use the data to finetune a small-scale LM (e.g., RoBERTa [12]), yielding a model that directly applies to and excels in diverse problems and domains.

We evaluate ALIGN with extensive experiments. **First**, we test on 25 seen and unseen datasets of the aforementioned tasks, and show our alignment model based on RoBERTa (355M parameters) achieves on par or even better performance than the FLAN-T5 models (780M and 3B) that are 2x or 8.5x as large and trained with substantially more data. In addition, the single alignment model outperforms RoBERTa specifically finetuned on each of the datasets. **Second**, we use ALIGN to evaluate the factual consistency of natural language generation (NLG) systems (e.g., for summarization, dialog, paraphrasing, etc.). On 23 datasets, the small alignment model achieves substantially higher correlation with human judgements than recent metrics, including those based on GPT-3.5 and even GPT-4. **Third**, we use ALIGN as a question answerability verifier and incorporate it as an add-on component to existing LLMs (e.g., GPT-3.5 and FLAN-T5). It significantly enhances the LLMs' performance in three question answering datasets, improving the average exact match score by 17.94 and F1 score by 15.05.

## 2 Related Work

Recent work has shown that LLMs are few-shot learners capable of generalizing to diverse tasks [13, 2, 14]. These LLMs are designed based on the principle of next-word-prediction, where the joint probability distribution of text sequences are factored into a product of conditional probabilities [15]. The performance of LLMs is highly correlated with their scales. Wei et al. [1] show that a scale of more than $10^{22}$ training FLOPs (around 10B model parameters) is required for several different LLM designs to achieve above-random performance on many language tasks.

Another line of research has tried to design models that can either learn from multiple tasks or handle many downstream tasks. Liu et al. [8] propose to use a BERT model [16] with task-specific heads to learn four types of tasks, including single-sentence classification, pairwise text classification, text similarity scoring, and relevance ranking. Aghajanyan et al. [9] *pre-finetune* language models on 50 dataset to encourage learning more general representations, and show that the process improves model performance and data-efficiency in the finetuning stage. Yin et al. [4], Wang et al. [5] and Mishra et al. [6] explore the application of a model trained with NLI datasets to multiple downstream tasks, in an effort to find more general yet still efficient methods. Some research explores the use of smaller models for enhancing LLMs. For example, Cappy [17], a small scorer model trained on the same datasets as in T0 [18], functions well on natural language classification tasks while boosting the performance of LLMs as an add-on. While we also use diverse datasets to train our model, we 1) unify language tasks into a single text pair alignment problem, 2) share all model components across multiple tasks and do not use dataset-specific heads, and 3) our model can be directly applied to a wide range of tasks without additional finetuning. The work demonstrates the strong potential of unified modeling and learning with all diverse relevant forms of experience [19].

Text alignment has long been used for measuring the correspondence of information between two pieces of text. For example, in machine translation, Brown et al. [20] propose a model that learns alignment between two languages. Papineni et al. [21] use n-gram overlap to compare translated sentences with their references. Gao et al. [22] train a sentence-level embedding model that compares similarity between two sentences while BERTScore [23] utilizes token-level alignment for evaluating text generation. Zha et al. [11] also propose building an automatic factual consistency metric for NLG systems through a text alignment framework. Expanding on the idea of text alignment, we explore how the formulation enables training a single alignment model that excels at a wide variety of tasks, including natural language understanding, factual consistency evaluation and answerability verification.

## 3 Text Alignment Model

In this section, we introduce the text pair alignment formulation. We first formally define the concept of text pair alignment, and then discuss how the alignment function can be used to solve a set of popular language tasks. Additionally, we cover the split-then-aggregate method used to handle long inputs. In Section 3.1, we discuss the training process of the alignment model (ALIGN).

Given a text pair $(x_1, x_2)$, we define text $x_2$ to be aligned with text $x_1$ if all information in $x_2$ is supported by information in $x_1$, following Zha et al. [11]. For example, let `"I have been in Kentucky, Kirby."` be text $x_1$. Then, `"I have been in the US."` is aligned with $x_1$. In contrast, both `"I have been in Europe."` and `"Kentucky has the best fried chicken."` are not aligned with $x_1$, as the former is contradicted by $x_1$, and the latter cannot be inferred from $x_1$. Formally, we model alignment as a function that maps the text pair $(x_1, x_2)$ to a label $y$ describing the level of alignment:

$$f : (x_1, x_2) \to y .$$

In practice, the language tasks we wish to solve with the alignment function can be broadly categorized into two groups: one that uses discrete labels, and the other that uses continuous labels (e.g., semantic textual similarity). More specifically, tasks with discrete labels are typically formulated as either binary classification (e.g., paraphrase detection) or three way classification (e.g., fact verification). In order to make the alignment function more general, such that it accommodates all the above cases, our alignment model produces three outputs: $\Pr(y_{\text{bin}})$, $\Pr(y_{\text{3way}})$, and $y_{\text{reg}}$. Here, $\Pr(y_{\text{bin}})$ and $\Pr(y_{\text{3way}})$ are probability distributions over the binary (ALIGNED, NOT-ALIGNED) and 3-way (ALIGNED, CONTRADICT, NEUTRAL) classification labels, respectively; $y_{\text{reg}} \in [0, 1]$ is a real-valued score for regression tasks.

This formulation allows us to apply the alignment function to diverse tasks:

- For tasks that naturally fit into the text pair alignment format, such as **NLI** [7], **fact verification** [24], **paraphrase detection** [25], and **semantic textual similarity** [26], depending on the nature of the task, we simply map one of the alignment labels to the desired label. For example, for most NLI tasks, we interpret the corresponding $y_{\text{3way}}$ labels as "entailment","contradiction", and "neutral".

- In **information retrieval** tasks [27], the goal is to find documents that can answer a given query from a large set of candidate documents. Since relevant documents contain information related to respective queries, we use candidate documents as text $x_1$, and queries as text $x_2$. Then, a higher $\Pr(y_{\text{bin}} = \text{ALIGNED})$ indicates the candidate document is more likely to be useful in answering the query.

- In **multiple choice QA** tasks [28], the inputs are a context, a question, and several choices (with one of them being the correct answer). In **extractive QA** tasks (including ones with **unanswerable questions** [29]), the inputs only consist of a context and a question. In either case, the expected output (the correct answer) can be inferred from the question and the context, while a wrong answer either contradicts the context or is not supported by the context. Therefore, we use the context as text $x_1$ and the concatenation of the question a candidate answer as text $x_2$. Here, a higher $\Pr(y_{\text{bin}} = \text{ALIGNED})$ indicates the candidate answer is more likely to be correct.

- In **coreference resolution** tasks [30], each sample includes a context containing a pronoun, and a list of candidate entities. The goal is to find the correct entity that the pronoun is referring to. As the pronoun and the associated entity is equivalent in this context, we consider the context with the pronoun replaced with the correct entity to be aligned with the original context. To solve coreference resolution problems, we simply replace the pronoun with candidate entities and compare the resulting contexts ($x_2$) with the original context ($x_1$). We pick the candidate that produces the highest $\Pr(y_{\text{3way}} = \text{ALIGNED})$ or $\Pr(y_{\text{bin}} = \text{ALIGNED})$ as the correct answer.

- For generative tasks like machine summarization, dialog, and paraphrasing, the alignment function can be used to **evaluate the factual consistency** of generated outputs. We use the generation context (e.g., input document) as text $x_1$, and candidate system output (e.g., generated summary) as text $x_2$. In this case, the probability of $\Pr(y_{\text{3way}} = \text{ALIGNED})$ or $\Pr(y_{\text{bin}} = \text{ALIGNED})$ indicates if the candidate output faithfully reflects information in the context, without introducing hallucinations or contradictions.

One specific challenge of applying the alignment function to downstream tasks is that text $x_1$ in some datasets (e.g., contexts in QA or summarization datasets) tends to be significantly longer than the input length limit of typical language models (e.g., 512 tokens for RoBERTa). As a result, naively truncating oversized inputs could throw away important information. To alleviate this problem, inspired by Laban et al. [31], Amplayo et al. [32], at inference time, instead of truncating the inputs, we split text $x_1$ into a set of chunks $\{x_1^{(i)}\}$ and text $x_2$ into a set of sentences $\{x_2^{(j)}\}$ such that the combined length of a chunk-sentence pair is slightly below that length limit. Then, we evaluate each pair and aggregate the results as

$$\text{alignment}(x_1, x_2) = \operatorname*{mean}_{j} \operatorname*{max}_{i} f(x_1^{(i)}, x_2^{(j)}) , \tag{1}$$

where the $\max$ operation selects the output with the highest ALIGNED probability or regression score. Since in most downstream applications, text $x_2$ tends to be succinct (e.g., summaries) and consists of self-contained sentences, this aggregation method can be interpreted as first finding the text $x_1$ chunk that most strongly supports each text $x_2$ "fact", and then taking the average across all text $x_2$ "facts".

## 3.1 Training

Our formulation not only allows us to solve the above tasks with a single alignment function, but also *learn* the alignment function from these tasks. By adapting text pair understanding tasks into a uniform alignment format as above, we can naturally model these tasks as simple classification and regression, allowing us to train a small model while achieving strong performance. Specifically, we use RoBERTa [12] as a lightweight backbone language model, and attach three individual linear layers to predict the three types of alignment outputs, $\Pr(y_{\text{bin}})$, $\Pr(y_{\text{3way}})$, and $y_{\text{reg}}$, respectively. The two classification heads are trained with cross entropy loss, while the regression head is trained with mean squared error loss. The losses are aggregated as a weighted sum,

$$\mathcal{L}_{\text{total}} = \lambda_1 \mathcal{L}_{\text{3way}} + \lambda_2 \mathcal{L}_{\text{bin}} + \lambda_3 \mathcal{L}_{\text{reg}} ,$$

where we set $\lambda_1 = 1/\log 3$, $\lambda_2 = 1/\log 2$, and $\lambda_3 = 1$, following Aghajanyan et al. [9].

Besides the aforementioned downstream datasets, we also include synthetic data to increase the diversity of the training set. Specifically, for QA datasets without wrong options (e.g., extractive QA datasets like SQuAD v2 [29]), we first remove the ground truth answer from the context, and then

Table 1: Performance of ALIGN and the much larger FLAN-T5 on in-domain datasets. **Bold** number indicates our performance is better than either FLAN-T5-large or xlarge.

| | | ALIGN (Ours) | | FLAN-T5 | |
|---|---|---|---|---|---|
| | | **base** | **large** | large | xlarge |
| Model Parameters | | **125M** | **355M** | 780M | 3B |
| NLI | MNLI-mm [7] | 87.5 | **90.3** | 88.7 | 90.4 |
| | MNLI-m [7] | 87.8 | **90.3** | 88.8 | 90.5 |
| | ANLI-1 [39] | 65.3 | **75.8** | 68.1 | 77.0 |
| | ANLI-2 [39] | **48.7** | **52.4** | 48.7 | 60.6 |
| | ANLI-3 [39] | 45.5 | **52.3** | 49.8 | 56.6 |
| | SNLI [40] | **90.8** | **91.8** | 89.7 | 90.7 |
| Fact Verification | NLI-FEVER [41, 39] | **76.8** | **77.8** | 72.0 | 71.9 |
| | VitaminC [24] | **89.8** | **91.8** | 72.9 | 73.9 |
| STS | SICK [26] | **90.7** | **91.5** | 79.3 | 79.1 |
| | STSB [42] | **89.0** | **89.8** | 83.9 | 88.2 |
| Paraphrase | PAWS [25] | 92.3 | 92.6 | 94.0 | 94.6 |
| | PAWS-QQP [25] | **91.9** | **93.8** | 88.3 | 90.1 |
| | QQP [43] | **90.1** | **91.3** | 86.8 | 87.4 |
| QA | RACE-m [28] | 76.9 | **86.8** | 84.8 | 87.6 |
| | RACE-h [28] | 68.8 | **81.6** | 78.3 | 84.6 |
| | Multi-RC [44] | 82.2 | **87.8** | 84.7 | 88.2 |
| | BoolQ [45] | 81.1 | **87.7** | 84.9 | 89.6 |
| | QuAIL [46] | 67.8 | 78.6 | 79.1 | 86.3 |
| | SciQ [47] | 92.4 | 93.7 | 94.9 | 95.7 |
| Coreference | GAP [30] | 81.4 | **88.6** | 73.8 | 81.5 |
| Average | | **79.8** | **84.3** | 79.6 | 83.2 |

use a QA model [13] to generate wrong answers that can be used to create NOT-ALIGNED samples. Additionally, we create synthetic paraphrase samples by back translating the WikiText-103 corpus [33] using a neural machine translation model [34]. For the WikiHow summarization dataset, we use an extractive summarizer [35] to generate synthetic summaries in additional to ground truth summaries. Following Kryscinski et al. [36], Deng et al. [37], we create negative samples for both WikiText-103 and WikiHow samples by randomly masking 25% of the tokens in text $b$ and infilling with a small masked language modeling model [38]. In total, we collect 5.9M examples from 28 datasets to train our alignment model ALIGN. We include more details of our training setup and data in Appendix C.

## 4   Experiments

In this section, we experiment with applying ALIGN to multiple downstream tasks, including language pair understanding tasks (Section 4.1), factual consistency evaluation (Section 4.2), and question answering with unanswerable questions (Section 4.3). We discuss experiment details and include a data contamination analysis in the Appendix (Section D).

### 4.1   Natural Language Understanding Tasks

Natural Language Understanding (NLU) is a major category of tasks for language models, and our formulation allows us to directly use ALIGN to solve these tasks. Specifically, we include NLI, fact verification, paraphrase detection, multiple-choice QA, STS, and coreference resolution datasets in the experiments. We also include unseen datasets to demonstrate the generalizability of ALIGN. Experiments show the alignment model is on par with FLAN T5 that has 8.5x as many parameters. Additionally, without further task-specific finetuning, our model outperforms finetuned language models of a similar size.

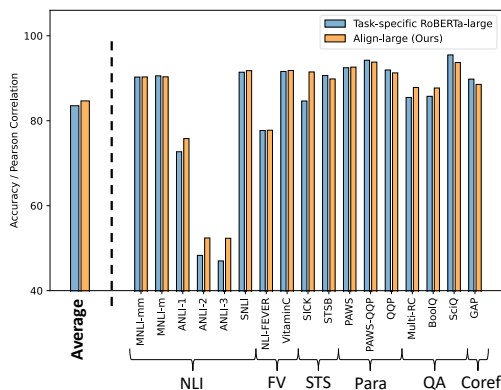
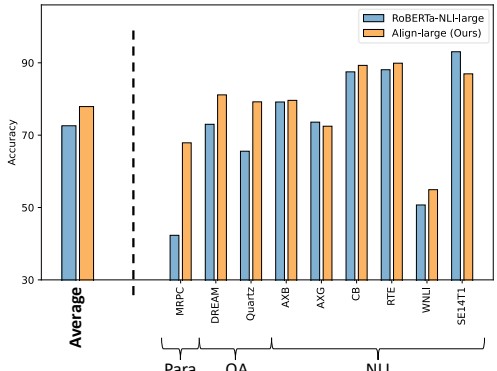

Figure 2: The performance of the ALIGN-large and the task-specific finetuned RoBERTa-large on in-domain tasks.

Figure 3: The performance of the ALIGN-large and RoBERTa-NLI-large on zero-shot tasks.

### 4.1.1 Experiment Setup

**Datasets** We first evaluate ALIGN on test sets of the 20 datasets used during training (in-domain setting; see Table 1). Then, we use 9 unseen datasets for evaluation (zero-shot setting; see Table 2). For more details about the datasets, please refer to appendix C.2. If a dataset does not have a public test set, we use its validation set instead. For datasets that require binary, 3-way classification or regression, we use the associated output heads, respectively, as discussed in Section 3.

**Baselines** To demonstrate the efficiency of ALIGN, we compare it with FLAN-T5 [14] and FLAN-Alpaca[3] with model size ranging from 220M (FLAN-Alpaca-base) to 3B (FLAN-Alpaca-xlarge and FLAN-T5-xlarge). For both models, we use the same prompts as in Longpre et al. [48]. We include task-specific RoBERTa models that are individually finetuned on each training set and evaluated on the corresponding test set to show that our alignment model works well out-of-the-box without further finetuning. We also compare with a multi-task RoBERTa model trained on all the original datasets (before converting to the alignment-format data), and a T5-base model instruction finetuned on our alignment datasets to show the effectiveness of our alignment formulation. Lastly, we compare with RoBERTa model finetuned on MNLI, ANLI and SNLI datasets (RoBERTa-NLI) in the zero-shot setting, to show the generalizability of our proposed formulation.

### 4.1.2 Results

We report average Pearson Correlation coefficient for the STS tasks [26, 42], and average accuracy for the other tasks.

In the **in-domain setting**, as show in Table 1, ALIGN outperforms FLAN-T5 that is 2x as large (780M) and has comparable performance to the version 8.5x as large (3B). Similar performance gain is also observed when comparing with FLAN-Alpaca in Table 9. Furthermore, ALIGN is on par with the task-specific finetuned RoBERTa models (Figure 2) and the multi-task RoBERTa model (Table 10). ALIGN outperforms the the instruction finetuned T5 model (Table 12), showing the effectiveness of our unified alignment formulation.

In the **zero-shot setting**, ALIGN achieves comparable performance with similarly sized variants of FLAN-T5 (Table 2), even on datasets that exist in FLAN-T5's training set. It also shows superiority to the multi-task RoBERTa in Table 11 as it eliminates the need to choose task heads while outperforming the average performance of the heads in the multi-task RoBERTa model. Additionally, ALIGN has stronger performance on average than the RoBERTa-NLI model (Figure 3) and the instruction finetuned T5 model (Table 13), indicating that our formulation leads to better generalizability.

---

[3] https://github.com/declare-lab/flan-alpaca

Table 2: Zero-shot setting results from ALIGN. The **bold** numbers indicates a better performance of ALIGN when comparing with similarly sized FLAN-T5. The gray number shows the specific dataset is appeared in the training set of FLAN-T5.

| | | **ALIGN (Ours)** | | FLAN-T5 | |
| --- | --- | --- | --- | --- | --- |
| | | **base (125M)** | **large (355M)** | base (250M) | large (780M) |
| NLI | AXB [49] | **75.1** | **79.6** | 71.7 | 76.2 |
| | AXG [49] | **59.8** | 72.5 | 53.4 | 73.6 |
| | CB [49] | 76.8 | **89.3** | 82.1 | 87.5 |
| | RTE [50] | **83.4** | **89.9** | 81.6 | 87.0 |
| | WNLI [50] | **52.1** | 54.9 | 46.5 | 62.0 |
| | SE14T1 [51] | **90.7** | **86.9** | 69.6 | 69.9 |
| Paraphrase | MRPC [52] | 66.0 | 67.9 | 74.8 | 80.1 |
| QA | DREAM [53] | **71.3** | **81.1** | 69.9 | 79.0 |
| | Quartz [54] | 59.7 | 79.2 | 74.4 | 90.2 |
| | Average | **70.5** | **77.9** | 69.3 | 78.4 |

## 4.2 Factual Consistency Evaluation for Language Generation

Studies have shown that natural generation systems (NLG) are prone to generating text that is not consistent with the source material [55–59]. As a result, many automatic metrics have been developed with the goal of detecting factual consistency errors. As factual consistency is closely related to our definition of text pair alignment, we can directly apply ALIGN for this purpose, using the NLG input context as $x_1$, and system outputs as $x_2$. We consider a system output with higher $\Pr(y_{3way} = \text{ALIGNED})$ to be more factually consistent.

### 4.2.1 Experiment Setup

**Dataset** Following Zha et al. [11], we use two popular factual consistency evaluation benchmarks, **TRUE** (containing 11 datasets, including dialog, fact verification, and paraphrase detection ) [59] and **SummaC** (consisting of 6 summarization datasets) [31]. We also include **Other** popular meta-evaluation datasets, namely XSumFaith [58], SummEval [60], QAGS-XSum [61], QAGS-CNNDM [61], FRANK [62] and SamSum [63]. This results in 23 datasets in total for our study.

**Baselines** We compare ALIGN with the latest LLM based automatic metrics: GPTScore [64], G-EVAL [65] and a ChatGPT-based metric [66]. These metrics achieve the best performance when using the GPT family of LLMs, which are significantly larger than our alignment model (e.g., GPT-3 has 175B parameters). GPTScore evaluates texts based on the probability of a LLM generating the target text, while G-EVAL augments its prompt using chain-of-thoughts techniques and asks the LLM to score the input by form-filling. Liu et al. [65] design a prompt that asks ChatGPT to score the faithfulness of the summary on a five point scale. Additionally, we include strong, smaller-scale (similar with our alignment model) baselines, including BERTScore [23], BLEURT [67], BARTScore [68], CTC [37], UniEval [69] and QAFactEval [70], following Zha et al. [11].

**Metrics** Both the TRUE and SummaC benchmarks formulates factual consistency evaluation as binary classification (i.e., identifying factual consistency errors). Following the common practice, we report ROC AUC [71], treating each model as a classifier. For the rest of datasets, we report instance-level Pearson, Spearman, and Kendall-$\tau$ correlation coefficients between automatic metric scores and human-annotated scores.

### 4.2.2 Results

For the LLMs-based metrics, we use the results reported by Fu et al. [64], Liu et al. [65], Gao et al. [66], and consequently results for some model-dataset combinations are unavailable. Despite being much smaller than ChatGPT/GPT-3.5 or GPT-4, our alignment model achieves comparable performance on SummEval (see Table 3). When evaluated on the QAGS-XSum and QAGS-CNNDM datasets, even our 125M alignment model outperforms both G-EVAL and GPTScore based on

GPT-3.5, while the 355M alignment model beats G-EVAL based on GPT-4. When compared with similarly sized metrics, our method consistently outperform the strong baselines on factual consistency benchmarks and datasets (see Figure 4). We include detailed results in Appendix D.

Table 3: The Spearman Correlation coefficient of ALIGN and GPT-based models on SummEval, QAGS-XSUM and QAGS-CNNDM datasets. **Bold** number shows the best model on a specific dataset.

|  |  | G-EVAL-3.5 GPT3.5-d03 | G-EVAL-4 GPT4 | GPTScore GPT3.5-d03 | ChatGPT GPT3.5-turbo | ALIGN-base (Ours) | ALIGN-large (Ours) |
|---|---|---|---|---|---|---|---|
| Model Parameters |  | — | — | — | — | **125M** | **355M** |
| Datasets | SummEval | 38.6 | **50.7** | 47.5 | 43.3 | 42.0 | 47.9 |
|  | QAGS-XSUM | 40.6 | 53.7 | 22.0 | — | 52.7 | **57.4** |
|  | QAGS-CNNDM | 51.6 | 68.5 | — | — | 56.1 | **71.6** |

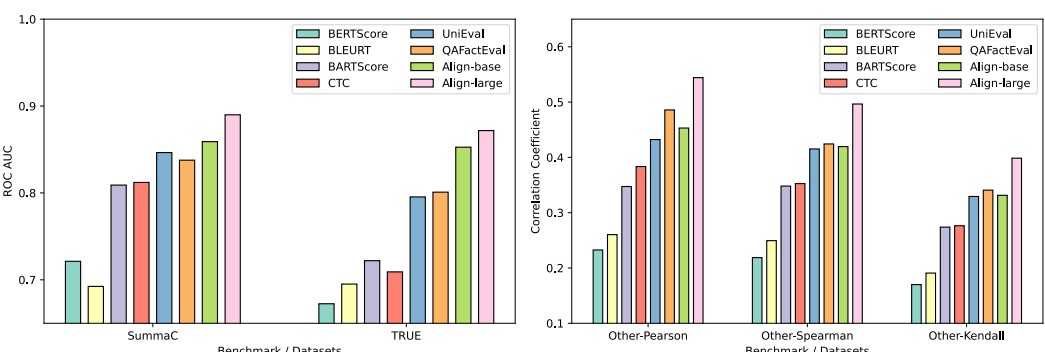

Figure 4: The performance of different models on diverse factual consistency benchmarks and datasets. The left figure includes performance on the SummaC and TRUE benchmark. The right figure shows models measured by correlation coefficients on other datasets (Section 4.2.1).

## 4.3 Question Answering with Unanswerable Question

In question answering tasks, a system must find the correct answer to a question from a context. When the question cannot be answered with information in the context, the system must indicate the question is not answerable. Despite being a well-studied task, predicting whether a question is answerable remains challenging, especially in a zero-shot setting.

A common approach to improve a system's ability to handle unanswerable questions is to introduce a verifier model in addition to the QA model [72, 73]. Given a context and question pair, the QA model first predicts a candidate answer. Then, the verifier model independently predicts whether the question is answerable by comparing the candidate answer to the question and the context. Lastly, the outputs of the two models are aggregated to form the final prediction. In our experiments, we use the alignment model as the verifier.

### 4.3.1 Experiment Setup

**Datasets** We experiment with two existing QA datasets with unanswerable questions, SQuAD v2 [29] and ACE-whQA [74]. Additionally, we construct a third dataset, Simplified Natural Questions (Simplified NQ), base on Natural Questions [75]. To build the dataset, for samples in Natural Questions with both short and long answers, we use the long answer as the context, and the short answer as the ground truth answer; for samples without short and long answers, we select random paragraphs from the articles as contexts and consider them to be unanswerable. Both ACE-whQA and Simplified NQ are not seen by the alignment model during training (i.e., a zero-shot experiment). We use the validation split of SQuAD v2 and Simplified NQ as their test splits are not publicly available.

**Baselines** We include FLAN T5 [14] and GPT-3.5[4] to represent large sequence-to-sequence language models. In addition, we experiment with using ALIGN as a verifier add-on for FLAN T5 and GPT-3.5. Here, we use $1 - \Pr(y_{bin} = \text{ALIGNED})$ as the unanswerable probability and use the SQuAD v2 validation split to find the best unanswerable threshold that maximizes the F1 score. The prompts we use and other experiment details are discussed in the appendix (Section D).

**Metrics** We follow Rajpurkar et al. [76] and report exact match and macro-averaged F1 score. To evaluate each model's performance at identifying unanswerable questions, we also formulate the problem as a binary classification task (predicting whether the sample is answerable) and report the ROC AUC [71]. A higher ROC AUC indicates the model is better at identifying unanswerable questions. For GPT-3.5 and FLAN T5, we consider the unanswerable classifier output to be 0 if the model predicts an answer, or 1 otherwise.

### 4.3.2 Results

As shown in Table 4, using ALIGN as a verifier add-on significantly improves GPT-3.5 and FLAN T5 in most cases (increases exact match score by 17.94 on average and F1 score by 15.05), suggesting that it is effective at identifying unanswerable questions. For the Simplified NQ dataset, adding the alignment verifier to GPT-3.5 degrades exact match and F1 score, but improves AUC. This indicates that while ALIGN produces meaningful unanswerable probabilities on the Simplified NQ dataset, the threshold found on the SQuAD v2 validation split is not ideal for Simplified NQ. Repeating the experiment with the best threshold selected on the Simplified NQ validation split (see numbers in parenthesis in Table 4) shows the potential for improvements in exact match and F1 scores, albeit this can no longer be considered a zero-shot setting.

Table 4: QA experiment results. Cases where adding the ALIGN verifier improves performance are highlighted in green. Best model for each dataset is shown in **bold**. For the combination of GPT-3.5 + Verifier and Simplified NQ, we also report the exact match and F1 scores with the best unanswerable threshold selected on the Simplified NQ validation split in parenthesis.

| | SQuAD v2 | | | ACE-whQA | | | Simplified NQ | | |
|---|---|---|---|---|---|---|---|---|---|
| | EM | F1 | AUC | EM | F1 | AUC | EM | F1 | AUC |
| GPT-3.5 | 52.53 | 63.96 | 0.76 | 67.98 | 71.98 | 0.77 | 58.37 | 68.61 | 0.81 |
| FLAN T5 | 75.72 | 79.01 | 0.83 | 26.29 | 29.24 | 0.51 | 38.24 | 44.98 | 0.58 |
| GPT-3.5 + Verifier (Ours) | 67.19 | 77.63 | 0.93 | **79.02** | **80.91** | 0.84 | 56.16 (63.51) | 57.40 (71.83) | **0.86** |
| FLAN T5 + Verifier (Ours) | **83.72** | **86.55** | **0.95** | 75.75 | 77.60 | **0.90** | **64.93** | **67.99** | 0.83 |

Table 5: Ablation results on in-domain natural language understanding tasks. Each row corresponds with a model trained on data adapted from incrementally more types (+) of tasks from scratch, or from fewer types (-) of tasks from the entire alignment training dataset. For example, the model on the second row is trained with NLI, Fact Verification and Paraphrase tasks. +QA (ALIGN-base) refers to the model trained on the same data as ALIGN-base. We report the average performance for each evaluation tasks. The last column shows the overall average for all the evaluated tasks. The best model for each evaluated task is shown in **bold**.

| | Evaluation Tasks | | | | | | |
|---|---|---|---|---|---|---|---|
| Training Tasks | NLI | Fact Verification | STS | Paraphrase | QA | Coreference | Average |
| +NLI | 69.0 | 65.3 | 53.7 | 69.3 | 56.0 | 70.6 | 64.0 |
| +FV, Para | 70.1 | 83.0 | 51.6 | **92.4** | 53.6 | 67.9 | 69.8 |
| +Coref, Sum, IR, STS | 69.9 | 82.9 | **90.3** | 91.9 | 51.8 | **83.2** | 78.3 |
| +QA (**ALIGN-base**) | **70.9** | **83.3** | 89.9 | 91.4 | 78.2 | 81.4 | 82.5 |
| -Synthetic | 70.4 | 83.1 | 90.1 | 92.0 | **78.6** | 83.1 | **82.9** |

---

[4]gpt-3.5-turbo, see https://platform.openai.com/docs/models/gpt-3-5

## 4.4 Ablation Study

As discussed in Section 3.1, ALIGN is trained on datasets from a wide set of language understand tasks. To understand their contributions to the performance of the alignment model, we conduct an ablation study by incrementally adding subsets of tasks to the training set. Specifically, we start with only NLI dataset, and then add the remaining tasks in the following order: 1) paraphrase detection (*para*) and fact verification (*FV*) datasets; 2) coreference resolution (*coref*), summarization (*sum*), information retrieval (*IR*), and STS datasets, and lastly 3) QA datasets. Additionally, we train an alignment model without synthetic data to measure the contribution of such data. For simplicity, we use RoBERTa-base as the backbone in this experiment. As shown in Table 5, each added subset improves the overall performance of the alignment model, suggesting our training tasks are compatible and contribute to the model performance. We notice that removing the synthetic data could slightly improve the overall performance, possibly due to the quality of the synthetic data. We will leave this for future study.

## 5    Conclusion

We propose to unify diverse language tasks into a text pair alignment problem. This framework yields an alignment model (ALIGN) that, despite being less versatile than LLMs, solves a wide range of language problems efficiently with superior performance. We show that ALIGN outperforms task-specific models finetuned on several NLU tasks while having performance comparable to LLMs that are orders of magnitude larger. Additionally, ALIGN excels in factual consistency evaluation, and can be used as an add-on to augment LLMs in QA tasks by identifying unanswerable questions.

**Limitations**   Our alignment framework uses splitting and aggregation to handle long inputs (see Section 3), with the assumption that text $x_2$ is short and its sentences are self-contained. While we empirically show this method works well on diverse datasets, violating this assumption has a few implications. First, if text $x_2$ sentences are highly interrelated, splitting them discards document-level semantic information, which could degrade performance. Second, as we need to evaluate all text $x_2$ sentences individually, doing so will be slow for long text $x_2$.

We use a wide collection of NLU datasets to learn the alignment function, with the assumption that these dataset, after being adapted into the text pair alignment format, accurately reflect our definition of alignment. However, as with all datasets, they could contain biases that are subsequently learned by our alignment model. Additionally, we augment the training set with synthetic data. While it proves to improve performance in our experiments, synthetic data likely do not perfectly model real-world data distributions.

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

# Appendix

## A   Ethics Statement

While our text pair alignment model achieves state-of-the-art performance on many downstream tasks, like all models, it does make mistakes. For example, when used for fact verification or factual consistency evaluation, it could misidentify *factually correct* statements as *incorrect* and vice versa. Additionally, as we use publicly available datasets to train the alignment model, it might have learned biases inherent to those datasets. Thus, one should proceed with caution when using ALIGN for purposes other than NLP research.

## B   Comparison with Other Model Types

We illustrate the major differences between our approach, LLMs, multitask learning models, and task-specific finetuned models in Table 6. Compared with LLMs, our alignment function is more efficient but less versatile. In contrast to task-specific finetuned models, the alignment function is more general and can handle more types of tasks. Unlike multitask learning models, we unify language tasks into a single text pair alignment problem, and share model components across multiple tasks (as apposed to using dataset-specific prediction heads). As a result, our alignment function can be directly applied to a wide range of tasks out-of-the-box, without any finetuning.

Table 6: Comparison between the alignment function and other types of language models.

| Type | Model Example | Efficient | Out of the box | General |
|---|---|---|---|---|
| LLM | T5, PALM, UL2, GPT | ✗ | ✔ | ✔ |
| Multitask learning | MT-DNN, MUPPET | ✔ | ✗ | ✔ |
| Task specific LM | Finetuned RoBERTa/BERT | ✔ | ✗ | ✗ |
| **Text pair alignment** | **ALIGN (Ours)** | ✔ | ✔ | ✔ |

## C   Training Details

### C.1   Trainig Setup

We choose RoBERTa [12] as the backbone for the alignment model. ALIGN-base and ALIGN-large are based on RoBERTa-base and RoBERTa-large, respectively. For the experiments in Section 4, we train ALIGN for 3 epochs with a batch size of 32, following common practice [12, 16]. Other hyperparameters are listed in Table 7. For the finetuned RoBERTa and RoBERTa-NLI model in Section 4.1, we set batch size to 16 and 32, respectively.

Table 7: The hyperparameters for training the alignment model.

| Hyperparameter | ALIGN-base | ALIGN-large |
|---|---|---|
| Parameters | 125M | 355M |
| Batch Size | 32 | 32 |
| Epochs | 3 | 3 |
| Optimizer | AdamW | AdamW |
| Learning Rate | 1e-5 | 1e-5 |
| Weight Decay | 0.1 | 0.1 |
| Adam $\epsilon$ | 1e-6 | 1e-6 |
| Warmup Ratio | 0.06 | 0.06 |
| Random Seed | 2022 | 2022 |
| GPU | 2×3090 | 4×A5000 |
| GPU Hours | 152h | 620h |

## C.2 Training Datasets

We collect datasets that falls into the scope of alignment as mentioned in Section 3. Table 8 lists the datasets we use for training the alignment model. The size of these datasets ranges from 4k samples to 5M. Most of the datasets are used for binary classification task except some NLI, fact verification and STS datasets.

We only use the first 500k samples in each dataset due to limited computation resource, which results in 5.9M training samples in total. During training, the samples are randomly sampled from the entire adapted training sets.

Table 8: The training datasets of ALIGN. Note due to resource constraints, we only use at most 500k samples from each dataset to train the alignment model.

| NLP Task | Dataset | Training Task | Sample Count |
|---|---|---|---|
| NLI | SNLI [40] | 3-way classification | 550k |
| | MultiNLI [7] | 3-way classification | 393k |
| | Adversarial NLI [39] | 3-way classification | 163k |
| | DocNLI [77] | binary classficiation | 942k |
| Fact Verification | NLI-style FEVER [41, 39] | 3-way classification | 208k |
| | VitaminC [24] | 3-way classification | 371k |
| Paraphrase | QQP [43] | binary classficiation | 364k |
| | PAWS-Wiki [25] | binary classficiation | 695k |
| | PAWS-QQP [25] | binary classficiation | 12k |
| | WikiText-103 [33] | binary classficiation | 8M |
| STS | SICK [78] | regression | 4k |
| | STSB [42] | regression | 6k |
| QA | SQuAD v2 [29] | binary classficiation | 130k |
| | RACE [28] | binary classficiation | 351k |
| | Adversarial QA [79] | binary classficiation | 60k |
| | BoolQ [45] | binary classficiation | 19k |
| | DROP [80] | binary classficiation | 155k |
| | MultiRC [44] | binary classficiation | 24k |
| | HotpotQA [81] | binary classficiation | 362k |
| | NewsQA [82] | binary classficiation | 161k |
| | QuAIL [46] | binary classficiation | 41k |
| | Quoref [83] | binary classficiation | 39k |
| | ROPES [84] | binary classficiation | 22k |
| | SciQ [85] | binary classficiation | 47k |
| | StrategyQA [86] | binary classficiation | 5k |
| Information Retrieval | MS MARCO [27] | binary classficiation | 5M |
| Summarization | WikiHow [87] | binary classficiation | 157k |
| Coreference | GAP [30] | binary classficiation | 4k |

# D  Additional Experiment Details

## D.1  Natural Language Understanding Tasks

**Prompts**   For FLAN models, we use the same prompts as Longpre et al. [48]. For datasets that do not appear in Longpre et al. [48], we use prompts of similar tasks. The prompt used for each dataset is listed below.

MNLI, NLI-FEVER, VitaminC:
```
"Premise:  {premise}\n\nHypothesis:  {hypothesis}\n\nDoes the premise
entail the hypothesis?\n\nA yes\nB it is not possible to tell\nC no"
```

ANLI:
```
"{context}\n\nBased on the paragraph above can we conclude that
\"{hypothesis}\"?\n\nA Yes\nB It's impossible to say\nC No"
```

SNLI:
```
"If \"{premise}\", does this mean that \"{hypothesis}\"?\n\nA yes\nB it is
not possible to tell\nC no"
```

SICK, STSB:
```
"{sentence1}\n{sentence2}\n\nRate the textual similarity of these two
sentences on a scale from 0 to 5, where 0 is \"no meaning overlap\" and
5 is \"means the same thing\".\n\nA 0\nB 1\nC 2\nD 3\nE 4\nF 5"
```

PAWS, PAWS-QQP:
```
"{sentence1}\n{sentence2}\n\nDo these sentences mean the same thing?\nA
no\nB yes"
```

QQP:
```
"{question1}\n{question2}\nWould you say that these questions are the
same?\nA no\nB yes"
```

RACE, QuAIL, SciQ:
```
"{fact}\n{question}\n\nA {option 1}\nB {option 2}\nC {option 3} ..."
```

Multi-RC:
```
"{paragraph}\n\nQuestion:  \"{question}\"\n\nResponse:
\"{response}\"\n\nDoes the response correctly answer the question?\n\nA
no\nB yes"
```

BoolQ:
```
"{text}\n\nCan we conclude that {question}?\n\nA no\nB yes"
```

GAP:
```
"Context:  {context}\n Given the context, which option is true?  \n\nA
{option 1}\nB {option 2}\nC {option 3} ..."
```

**Multitask RoBERTa Baseline**    To obtain the multitask-learning RoBERTa model, we follow the popular multi-task learning work [9], and train the same base model with the same set of tasks (datasets) as our alignment model. Notably, different from our alignment model that uses a unified interface to accommodate all diverse tasks, the conventional multitask-learning model learns separate prediction heads for different tasks.

**Instruction Finetuned T5 Baseline**    In order to understand the performance difference between our text alignment formulation and instruction fine-tuning, we instruction-finetune T5-base (250M parameters) on the same datasets as our alignment model. We do not convert QA tasks since T5 naturally supports sequence generation. We follow the prompts mentioned in Chung et al. [14], Longpre et al. [48] and format the datasets to train the T5 model.

**Results**    We show the performance of finetuned RoBERTa and FLAN-Alpaca on these datasets in Table 9, while the result of the multi-task RoBERTA is in Table 10 and 11. The comparisons with the T5 model instruction finetuned on alignment datasets are in Table 12 and 13. We have compared ALIGN with finetuned RoBERTa on these datasets in Figure 2. When comparing FLAN-T5 and FLAN-Alpaca, we notice FLAN-T5 consistently outperforms FLAN-Alpaca on all scales. Thus, we compare our alignment model with FLAN-T5 in Table 1.

## D.2    Factual Consistency Evaluation for Language Generation

**Detailed Result**    In this section, we report the detailed results associated with Figure 4. We list the performance of each metric on SummaC Benchmark and TRUE Benchmark in Table 15 and Table 16, respectively. We also show the Pearson Correlation, Spearman Correlation and Kendall's tau Correlation coefficients on other datasets in Table 17, 18 and 19, respectively.

Table 9: The performance of finetuned RoBERTa and FLAN-Alpaca on the in-domain datasets. We report the average performance of each model and we also include the average without RACE and QuAIL.

| | | Finetuned RoBERTa | | FLAN-Alpaca | | |
| --- | --- | --- | --- | --- | --- | --- |
| | | base | large | base | large | xlarge |
| Model Parameters | | 125M | 355M | 220M | 770M | 3B |
| NLI | MNLI-mm | 87.2 | 90.3 | 79.9 | 86.4 | 89.3 |
| | MNLI-m | 87.9 | 90.6 | 80.0 | 87.2 | 89.4 |
| | ANLI-1 | 62.8 | 72.7 | 47.4 | 65.7 | 74.8 |
| | ANLI-2 | 44.5 | 48.3 | 38.2 | 46.6 | 57.6 |
| | ANLI-3 | 42.8 | 47.0 | 37.7 | 46.4 | 54.6 |
| | SNLI | 91.0 | 91.4 | 82.9 | 88.1 | 90.2 |
| Fact Verification | NLI-FEVER | 76.1 | 77.7 | 69.6 | 73.0 | 72.1 |
| | VitaminC | 89.3 | 91.6 | 63.3 | 72.5 | 77.4 |
| STS | SICK | 88.9 | 84.7 | 37.7 | 66.4 | 70.1 |
| | STSB | 89.8 | 90.6 | 33.4 | 52.5 | 79.5 |
| Paraphrase | PAWS | 92.3 | 92.5 | 68.1 | 92.0 | 93.0 |
| | PAWS-QQP | 94.7 | 94.2 | 57.6 | 85.1 | 87.1 |
| | QQP | 91.1 | 92.0 | 75.5 | 81.6 | 86.5 |
| QA | RACE-m | 74.6 | 24.0 | 64.3 | 78.5 | 87.8 |
| | RACE-h | 67.8 | 23.9 | 57.3 | 71.9 | 82.9 |
| | Multi-RC | 77.5 | 85.5 | 64.2 | 84.3 | 87.1 |
| | BoolQ | 79.1 | 85.7 | 71.7 | 82.0 | 87.2 |
| | QuAIL | 57.7 | 27.0 | 56.7 | 78.2 | 84.2 |
| | SciQ | 93.4 | 95.5 | 90.8 | 83.1 | 95.6 |
| Coreference | GAP | 74.3 | 89.8 | 58.4 | 65.6 | 80.7 |
| Average | | 78.1 | 74.7 | 61.7 | 74.4 | 81.4 |
| Average w/o RACE, QuAIL | | 80.2 | 83.5 | 62.1 | 74.0 | 80.7 |

**Ablation Study**   In Equation 1, we propose to aggregate chunk-sentence scores by taking the maximum score over context chunks and then average over claim sentences (mean-max). Another reasonable choice is to take the *minimum* score over claim sentences instead of the average (min-max). We evaluate the two aggregation methods on factual consistency evaluation tasks and report the results in Table 20. Overall, the mean-max method leads to better correlation with human annotation. We speculate this is because taking the average helps remove some of the noise introduced by the alignment estimator, while the min-max setup would be easily affected by underestimation.

### D.3   Question Answering with Unanswerable Question

**Simplified Natural Questions**   For this experiment, we construct a new SQuAD-style QA dataset with unanswerable questions, Simplified Natural Questions (Simplified NQ), base on Natural Questions [75]. For each sample (an search query and a Wikipedia article) in Natural Questions, [75] ask five annotators to find 1) an HTML bounding box (the *long answer*; e.g., paragraphs, tables, list items, or whole list) containing enough information to infer the answer to the query (long answer), and 2) a short span within the long answer that answers the question (the *short answer*). For both short and long answers, the annotators can alternatively indicate that an answer could not be found.

For the purpose of constructing Simplified NQ, we consider a sample to be answerable if at least 2 annotators identified both long and short answers. In this case, we use the most popular (among the annotators) short answer as the ground truth answer, and the most popular long answer containing the selected short answer as the context. Conversely, if less than 2 annotators identified any long answer, and less than 2 annotators identified any short answer, we consider the sample to be unanswerable and use a random paragraph from the article as the context. We discard all remaining samples to

Table 10: The performance of multi-task RoBERTa-base and ALIGN-base on the in-domain datasets. The last row is the averaged performance of each model on these datasets.

| Task | Dataset | Multitask-base | ALIGN-base |
|---|---|---|---|
| NLI | MNLI-mm | 87.61 | 87.54 |
| | MNLI-m | 87.68 | 87.82 |
| | ANLI-1 | 65.10 | 65.30 |
| | ANLI-2 | 48.20 | 48.70 |
| | ANLI-3 | 46.17 | 45.50 |
| | SNLI | 91.33 | 90.78 |
| Fact Verification | NLI-FEVER | 76.50 | 76.78 |
| | VitaminC | 89.82 | 89.79 |
| STS | SICK | 89.01 | 90.71 |
| | STSB | 87.86 | 89.03 |
| Paraphrase | PAWS | 93.94 | 92.33 |
| | PAWS-QQP | 92.32 | 91.88 |
| | QQP | 90.66 | 90.07 |
| QA | RACE-m | 78.34 | 76.95 |
| | RACE-h | 71.56 | 68.84 |
| | Multi-RC | 83.83 | 82.20 |
| | BoolQ | 81.32 | 81.07 |
| | SciQ | 92.10 | 92.40 |
| Coreference | GAP | 81.65 | 81.35 |
| Average | | 80.79 | 80.48 |

Table 11: The performance of multi-task RoBERTa-base and ALIGN-base on the zero-shot datasets. Multi-task RoBERTa uses task-specific prediction heads for each of its training sets. To test it on zero-shot datasets, we evaluate a set of "reasonable" heads obtained during training (e.g., we use heads trained on NLI datasets for NLI evaluation tasks) and report the average (avg. of heads), best (best head), and worst (worst head) performance of these heads. "Best head" assumes oracle access to evaluation results, and is thus effectively an unrealistic upper bound. The last row is the averaged performance of each configuration on these datasets.

| Task | Dataset | Multitask-base (avg. of heads) | Multitask-base (best head) | Multitask-base (worst head) | ALIGN-base |
|---|---|---|---|---|---|
| NLI | AXB | 74.94 | 76.18 | 72.37 | 75.09 |
| | AXG | 63.58 | 65.73 | 60.39 | 59.83 |
| | CB | 80.00 | 83.93 | 76.79 | 76.79 |
| | RTE | 81.41 | 81.95 | 81.23 | 83.39 |
| | WNLI | 56.34 | 60.56 | 47.89 | 52.11 |
| | SE14T1 | 57.86 | 58.45 | 57.17 | 90.72 |
| Paraphrase | MRPC | 69.60 | 71.19 | 68.23 | 65.97 |
| QA | DREAM | 67.04 | 73.98 | 57.03 | 71.34 |
| | Quartz | 59.09 | 63.52 | 54.46 | 59.69 |
| Average | | 67.76 | 70.61 | 63.95 | 70.55 |

Table 12: The performance of ALIGN-base and the instruction finetuned T5-base (IF-T5-base) on the in-domain tasks. The last column shows the average performance on the in-domain tasks.

| Model (Size) | NLI | Fact Verification | STS | Paraphrase | QA | Coreference | Average |
|---|---|---|---|---|---|---|---|
| ALIGN-base (125M) | 70.9 | 83.3 | 89.9 | 91.4 | 78.2 | 81.4 | 82.5 |
| IF-T5-base (222M) | 53.7 | 68.7 | 62.0 | 83.2 | 30.3 | 42.9 | 56.8 |

Table 13: The performance of ALIGN-base and the instruction finetuned T5-base (IF-T5-base) on the zero-shot tasks. The last row shows the average performance on the in-domain tasks.

| Dataset | Alignment-base | IF-T5-base |
|---------|----------------|------------|
| AXB | 75.1 | 65.3 |
| AXG | 59.8 | 50.8 |
| CB | 76.8 | 58.9 |
| RTE | 83.4 | 59.9 |
| WNLI | 52.1 | 47.9 |
| SE14T1 | 90.7 | 48.5 |
| MRPC | 66.0 | 66.1 |
| DREAM | 71.3 | 35.8 |
| Quartz | 59.7 | 50.6 |
| AVG | 70.5 | 53.8 |

Table 14: The performance of RoBERTa-NLI and FLAN-Alpaca on the zero-shot datasets (of alignment model). The gray number shows the specific dataset is appeared in the training set of FLAN-T5. We report the average performance of each model in the last row.

| | | RoBERTa-NLI | | FLAN-Alpaca | | |
|---|---|---|---|---|---|---|
| | | base | large | base | large | xlarge |
| Model Parameters | | 125M | 355M | 220M | 770M | 3B |
| NLI | AXB | 75.2 | 79.2 | 53.6 | 72.3 | 77.2 |
| | AXG | 59.6 | 73.6 | 49.4 | 72.5 | 88.8 |
| | CB | 85.7 | 87.5 | 78.6 | 78.6 | 87.5 |
| | RTE | 81.2 | 88.1 | 72.9 | 79.8 | 87.0 |
| | WNLI | 52.1 | 50.7 | 40.8 | 62.0 | 71.8 |
| | SE14T1 | 91.2 | 93.1 | 65.0 | 72.4 | 77.3 |
| Paraphrase | MRPC | 38.7 | 42.3 | 66.7 | 75.4 | 83.1 |
| QA | DREAM | 63.9 | 73.0 | 63.5 | 76.8 | 89.5 |
| | Quartz | 54.6 | 65.6 | 68.1 | 87.4 | 90.2 |
| Average | | 66.9 | 72.6 | 62.1 | 75.2 | 83.6 |

Table 15: The ROC AUC of each metric on the SummaC benchmark. CGS and XSF are abbreviations of CogenSumm and XSumFaith, respectively. The strongest performance on each dataset is shown in **bold**. The last column shows the average performance on each dataset in the SummaC benchmark.

| | SummaC Benchmark | | | | | | |
|---|---|---|---|---|---|---|---|
| | CGS | XSF | PolyTope | FactCC | SummEval | FRANK | AVG |
| BERTScore | 63.1 | 49.0 | 85.3 | 70.9 | 79.6 | 84.9 | 72.1 |
| BLEURT | 60.8 | 64.7 | 76.7 | 59.7 | 71.1 | 82.5 | 69.2 |
| BARTScore | 74.3 | 62.6 | 91.7 | 82.3 | 85.9 | 88.5 | 80.9 |
| CTC | 76.5 | 65.9 | 89.5 | 82.6 | 85.6 | 87.3 | 81.2 |
| UniEval | 84.7 | 65.5 | **93.4** | 89.9 | 86.3 | 88.0 | 84.6 |
| QAFactEval | 83.4 | 66.1 | 86.4 | 89.2 | 88.1 | 89.4 | 83.8 |
| **ALIGN-base (Ours)** | 80.6 | **76.1** | 87.5 | 93.1 | 88.6 | 89.5 | 85.9 |
| **ALIGN-large (Ours)** | **88.4** | 74.6 | 92.5 | **94.9** | **92.3** | **91.3** | **89.0** |

Table 16: The ROC AUC of each metric on the TRUE benchmark. The datasets with asterisks(*) appear in the training set of the alignment model. We compute both the overall average on all datasets (`Average`) and average without PAWS, FEVER, VitaminC datasets (`Average-ZS`). The latter shows the zero-shot performance of ALIGN. **Bold** indicates the best performance on a dataset.

| | | BERTScore | BLEURT | BARTScore | CTC | UniEval | QAFactEval | ALIGN-base (Ours) | ALIGN-large (Ours) |
|---|---|---|---|---|---|---|---|---|---|
| | FRANK | 84.0 | 81.6 | 87.8 | 87.1 | 88.1 | 88.5 | 90.4 | **91.4** |
| | SummEval | 72.3 | 68.0 | 78.9 | 79.8 | 81.2 | 80.9 | 79.5 | **83.8** |
| | MNBM | 52.5 | 65.5 | 63.5 | 65.0 | 66.8 | 67.3 | **76.4** | 74.4 |
| | QAGS-C | 70.6 | 71.2 | 83.9 | 77.3 | 86.5 | 83.9 | 80.3 | **89.0** |
| | QAGS-X | 44.3 | 56.2 | 60.2 | 67.7 | 76.7 | 76.1 | 79.6 | **83.2** |
| | BEGIN | 86.4 | 86.6 | **86.7** | 72.0 | 73.6 | 81.0 | 81.3 | 81.1 |
| TRUE | Q2 | 70.2 | 72.9 | 65.1 | 66.8 | 70.4 | 75.8 | 77.2 | **79.2** |
| Benchmark | DialFact | 68.6 | 73.0 | 60.8 | 63.7 | 80.4 | 81.8 | 83.3 | **85.1** |
| | PAWS* | 78.6 | 68.4 | 77.1 | 63.1 | 80.1 | 86.1 | 97.6 | **98.4** |
| | FEVER* | 54.2 | 59.5 | 66.1 | 72.5 | 92.1 | 86.0 | 94.7 | **94.9** |
| | VitaminC* | 58.2 | 61.8 | 64.2 | 65.0 | 79.1 | 73.6 | 97.8 | **98.3** |
| | Average | 67.2 | 69.5 | 72.2 | 70.9 | 79.5 | 80.1 | 85.3 | **87.2** |
| | Average-ZS | 68.6 | 71.9 | 73.4 | 72.4 | 78.0 | 79.4 | 81.0 | **83.4** |

Table 17: The Pearson correlation coefficients of various metrics on `other` datasets mentioned in Section 4.2.1. Q-XSum and Q-CNNDM are abbreviations of QAGS-XSum and QAGS-CNNDM, respectively. F-XSum and F-CNNDM are abbreviations of FRANK-XSum and FRANK-CNNDM, respectively. The last column shows the average performance on each dataset. The best performance is shown in **bold**.

| | Other Datasets - Pearson | | | | | | | |
|---|---|---|---|---|---|---|---|---|
| | XSumFaith | SummEval | Q-Xsum | Q-CNNDM | F-Xsum | F-CNNDM | SamSum | AVG |
| BERTScore | 13.0 | 33.1 | -10.6 | 51.7 | 13.0 | 51.7 | 10.9 | 23.3 |
| BLEURT | **38.7** | 23.8 | 13.2 | 45.2 | 15.6 | 37.5 | 8.1 | 26.0 |
| BARTScore | 29.3 | 35.5 | 16.3 | 71.5 | 23.7 | 51.9 | 15.0 | 34.7 |
| CTC | 27.2 | 54.7 | 30.6 | 64.5 | 20.0 | 54.5 | 16.9 | 38.3 |
| UniEval | 23.9 | 57.8 | 45.5 | 66.7 | 27.2 | 58.3 | 23.2 | 43.2 |
| QAFactEval | 30.3 | 61.6 | 44.2 | 68.4 | 32.1 | 64.6 | 38.9 | 48.6 |
| **ALIGN-base (Ours)** | 33.2 | 57.8 | 51.1 | 60.9 | 31.2 | 61.8 | 21.1 | 45.3 |
| **ALIGN-large (Ours)** | 28.8 | **66.7** | **53.9** | **76.1** | **38.9** | **68.9** | **47.7** | **54.4** |

Table 18: The Pearson correlation coefficients of various metrics on `other` datasets mentioned in Section 4.2.1. The format in this table follows Table 17.

| | Other Datasets - Spearman | | | | | | | |
|---|---|---|---|---|---|---|---|---|
| | XSumFaith | SummEval | Q-Xsum | Q-CNNDM | F-Xsum | F-CNNDM | SamSum | AVG |
| BERTScore | 13.4 | 31.5 | -8.9 | 46.2 | 12.7 | 45.1 | 13.1 | 21.9 |
| BLEURT | 37.0 | 23.6 | 12.4 | 43.4 | 13.9 | 37.6 | 6.7 | 24.9 |
| BARTScore | 29.8 | 39.1 | 17.0 | 68.1 | 20.0 | 53.3 | 16.3 | 34.8 |
| CTC | 29.8 | 41.7 | 30.6 | 57.3 | 20.4 | 49.4 | 17.7 | 35.3 |
| UniEval | 25.3 | 44.3 | 50.0 | 67.6 | 26.7 | 54.0 | 22.8 | 41.5 |
| QAFactEval | 31.9 | 42.8 | 44.1 | 63.1 | 25.5 | 53.7 | 35.9 | 42.4 |
| **ALIGN-base (Ours)** | **38.8** | 42.0 | 52.7 | 56.1 | 25.5 | 56.4 | 22.3 | 42.0 |
| **ALIGN-large (Ours)** | 32.1 | **47.9** | **57.4** | **71.6** | **30.0** | **61.8** | **46.7** | **49.7** |

Table 19: The Kendall's tau correlation coefficients of various metrics on `other` datasets mentioned in Section 4.2.1. The format in this table follows Table 17.

| | Other Datasets - Kendall's tau | | | | | | | |
| | XSumFaith | SummEval | Q-Xsum | Q-CNNDM | F-Xsum | F-CNNDM | SamSum | AVG |
|---|---|---|---|---|---|---|---|---|
| BERTScore | 9.2 | 24.9 | -7.3 | 36.3 | 10.4 | 34.7 | 10.7 | 17.0 |
| BLEURT | 25.3 | 18.6 | 10.1 | 33.9 | 11.4 | 28.8 | 5.5 | 19.1 |
| BARTScore | 20.2 | 31.0 | 13.9 | 55.6 | 16.3 | 41.4 | 13.3 | 27.4 |
| CTC | 20.2 | 33.2 | 25.1 | 45.7 | 16.6 | 38.2 | 14.4 | 27.6 |
| UniEval | 17.0 | 35.3 | 40.9 | 54.4 | 21.8 | 42.4 | 18.7 | 32.9 |
| QAFactEval | 23.2 | 34.0 | 36.2 | 50.5 | 22.4 | 42.2 | 30.1 | 34.1 |
| **ALIGN-base (Ours)** | **26.6** | 33.4 | 43.1 | 45.5 | 20.8 | 44.4 | 18.2 | 33.1 |
| **ALIGN-large (Ours)** | 21.9 | **38.4** | **47.0** | **59.6** | **24.5** | **49.5** | **38.2** | **39.9** |

Table 20: The comparison between the mean aggregation method and the min aggregation method. The **bold** number represents a better performance when comparing the aggregation method.

| | ALIGN-base mean-max | ALIGN-base min-max | ALIGN-large mean-max | ALIGN-large min-max |
|---|---|---|---|---|
| SummaC | 85.9 | 86.1 | 89.0 | 89.1 |
| TRUE | **85.3** | 84.2 | **87.2** | 86.2 |
| Other-Pearson | **45.3** | 43.7 | **54.4** | 52.1 |
| Other-Spearman | 42.0 | **42.3** | **49.7** | 49.2 |
| Other-Kendall | 33.1 | 33.1 | **39.9** | 38.8 |

avoid ambiguity (e.g., some samples might only have long answers but not short answers). This results in a total of 3336 answerable samples and 3222 unanswerable samples in the validation set.

**Prompts and QA Inference**    For FLAN T5, we follow [48] and use the following prompt:
```
Context:  {context}\nQuestion:  {question}\nAnswer:
```

For GPT-3.5, we use a prompt with additional instructions:
```
Find the answer to the question from the given context.  When the question
cannot be answered with the given context, say "unanswerable".  Just say
the answer without repeating the question.\nContext:  {context}\nQuestion:
{question}\nAnswer:
```

At inference time, we truncate the context if necessary such that the entire input is at most around 2000 tokens long (2000 for FLAN T5, 2040 for GPT-3.5 to account for the longer prompt). We use greedy decoding for FLAN T5, a the default chat completion settings for GPT-3.5. When FLAN T5 outputs "unanswerable", we interpret it as predicting the sample to be not answerable. Similarly, if GPT-3.5's output contains any of "unanswerable", "no answer", "context does not provide an answer", we consider the prediction to be unanswerable.

**Additional Results**    In addition to FLAN T5 and GPT-3.5, we also experiment with Electra [88], one of the top performing single models on the SQuAD v2 leaderboard, for reference. Specifically, we reproduce Clark et al. [88]'s design that use a QA prediction head to jointly predict the answer span and unanswerable probability. As shown in Table 21, while Electra is a strong performer on SQuAD v2 and Simplified NQ, adding the alignment verifier to GPT-3.5 and FLAN T5 greatly reduces the performance gap. Additionally, on ACE-whQA, our design (both FLAN T5 and GPT-3.5 with alignment verifiers) outperforms Electra.

## D.4  Ablation Study

We present the additional ablation result on factual consistency evaluation tasks in Table 22. This part follows Section 4.4, where we use the same checkpoints that are trained on incrementally added tasks. Result shows the training tasks are generally compatible and effective, though we notice adding fact verification and paraphrase detection tasks lead to a slightly performance drop. We speculate it is due to the paraphrase detection task, where a text pair is expected to have exactly the same information.

Table 21: Additional experiment results on QA with unanswerable questions including Electra. The best model for each task/metric is shown in **bold**.

| | SQuAD v2 | | | ACE-whQA | | | Simplified NQ | | |
|---|---|---|---|---|---|---|---|---|---|
| | EM | F1 | AUC | EM | F1 | AUC | EM | F1 | AUC |
| Electra | **86.47** | **89.37** | **0.97** | 52.32 | 55.59 | 0.87 | **70.81** | **74.13** | **0.88** |
| GPT-3.5 | 52.53 | 63.96 | 0.76 | 67.98 | 71.98 | 0.77 | 58.37 | 68.61 | 0.81 |
| Flan T5 | 75.72 | 79.01 | 0.83 | 26.29 | 29.24 | 0.51 | 38.24 | 44.98 | 0.58 |
| GPT-3.5 + Verifier (Ours) | 67.19 | 77.63 | 0.93 | **79.02** | **80.91** | 0.84 | 56.16 | 57.40 | 0.86 |
| FLAN T5 + Verifier (Ours) | 83.72 | 86.55 | 0.95 | 75.75 | 77.60 | **0.90** | 64.93 | 67.99 | 0.83 |

The ALIGN-base model, which uses all the possible training data, gets the best performance on every factual consistency evaluation task.

Table 22: Ablation results on factual consistency evaluation tasks. Each row corresponds with a model trained with data adapted from incrementally more types of tasks. For example, the model on the second row is trained with NLI, Fact Verification and Paraphrase tasks. The model on the last row is the same as Alignment-base. We report the average performance for each evaluation tasks. The last column shows the overall average for the factual consistency evaluation tasks. The best performance for each task is shown in **bold**.

| | Factual Consistency Evaluation Tasks | | | | | |
|---|---|---|---|---|---|---|
| Training Tasks | SummaC | TRUE | Other-Pearson | Other-Spearman | Other-Kendall | Average |
| +NLI | 78.1 | 77.5 | 32.6 | 33.6 | 26.3 | 49.6 |
| +FV, Para | 74.9 | 80.3 | 27.6 | 27.2 | 21.1 | 46.2 |
| +Coref, Sum, IR, STS | 84.2 | 83.7 | 39.4 | 36.8 | 28.8 | 54.6 |
| +QA (Alignment-base) | **85.9** | **85.3** | **45.3** | **42.0** | **33.1** | **58.3** |

## D.5 Evaluation Data Contamination Analysis

As we train and evaluate our models by adapting existing datasets, a subset of our evaluation data could be contaminated with training data. To understand the impact of data contamination, we perform a post hoc overlap analysis. Following [2], for each evaluation example, we check if any of its n-grams exist in the training set, where n is the 5th percentile example length for the evaluation dataset in words (in practice we clamp n to between 8 and 13 to avoid spurious collisions and limit run time). We consider an evaluation example "dirty" if there is at least one overlapping n-gram or "clean" otherwise. Due to resource constraints (e.g., available RAM), we sample up to 1000 examples from each NLU (Section 4.1) and factual consistency evaluation (Section 4.2) dataset for analysis.

The results are shown in Table 23. Overall, the SummaC benchmark and the other datasets we use in the factual consistency evaluation experiment have the least number of dirty examples, with other datasets having varying level of overlap with training data. One notable case is ANLI, where almost all examples are marked as dirty. We manually examine a small selection of ANLI examples, and find that most of the overlapping n-grams come from the premise portion of the examples (they overlap with examples from DocNLI and HotpotQA in the training set). As noted by [2], this n-gram overlap metric tends to show a large number of false positives for datasets constructed from common data sources. For instance, one possible explanation for a high dirty count is that training and evaluation examples share texts sourced from the Internet (e.g., Wikipedia) as supporting information (e.g., HotpotQA contexts and ANLI premises), which does not necessarily leak evaluation set answers. Thus, a high number of dirty examples alone does not meaningfully indicate contamination.

A more reliable indicator of data contamination is the difference in metric scores when evaluating using the clean subset compared to the full datasets. If removing the dirty samples (i.e., using the clean subset) leads to significantly worse metric scores, then the model might have overfit to the overlapping training data. As show in Table 23, NLI-FEVER and RACE-h have the biggest drop in metric scores. However, overall, removing dirty examples does not meaningfully reduce metric scores (the average difference is -0.1). Thus, the data contamination does not have a significant impact on our evaluation results and conclusions.

Table 23: Data contamination analysis results for evaluation datasets. For each evaluation dataset, we report the percentage of examples containing n-grams that also exist in training set ("dirty %"); the performance of ALIGN-large on the entire dataset ("full eval set"; see Section 4 for details), the clean subset, and the dirty subset; and lastly, the difference between the clean subset and the entire dataset ("Clean vs. Full"). We exclude subsets with less than 100 examples to reduce variance.

| Experiment | Dataset | Dirty % | Accuracy / Pearson Correlation / AUC | | | |
|---|---|---|---|---|---|---|
| | | | Full Eval Set | Clean Subset | Dirty Subset | Clean vs. Full |
| Language Understanding In-Domain | MNLI-mm | 2.8 | 90.3 | 90.0 | — | -0.3 |
| | MNLI-m | 5.4 | 90.3 | 88.6 | — | -1.8 |
| | ANLI-1 | 99.1 | 75.8 | — | 75.9 | — |
| | ANLI-2 | 99.4 | 52.4 | — | 52.3 | — |
| | ANLI-3 | 99.9 | 52.3 | — | 51.7 | — |
| | SNLI | 0.5 | 91.8 | 91.4 | — | -0.4 |
| | NLI-FEVER | 68.9 | 77.8 | 73.6 | 79.1 | -4.1 |
| | VitaminC | 18.5 | 91.8 | 90.4 | 94.6 | -1.4 |
| | SICK | 52.4 | 91.5 | 93.2 | 90.0 | 1.7 |
| | STSB | 12.2 | 89.8 | 88.8 | 91.3 | -1.0 |
| | PAWS | 16.5 | 92.6 | 91.7 | 92.7 | -0.9 |
| | PAWS-QQP | 78.4 | 93.8 | 97.3 | 92.8 | 3.5 |
| | QQP | 34.1 | 91.3 | 93.2 | 91.8 | 1.9 |
| | RACE-m | 34.0 | 86.8 | 88.0 | 82.6 | 1.3 |
| | RACE-h | 28.8 | 81.6 | 78.7 | 83.7 | -3.0 |
| | Multi-RC | 44.0 | 87.8 | 87.0 | 85.7 | -0.9 |
| | BoolQ | 49.5 | 87.7 | 85.9 | 90.9 | -1.8 |
| | Quail | 0.4 | 78.6 | 78.5 | — | -0.1 |
| | SciQ | 20.1 | 93.7 | 92.4 | 99.0 | -1.3 |
| | GAP | 1.6 | 88.6 | 87.5 | — | -1.1 |
| Language Understanding Zero-Shot | AXB | 1.2 | 79.6 | 79.8 | — | 0.1 |
| | AXG | 0.0 | 72.5 | 72.5 | — | 0.0 |
| | CB | 12.5 | 89.3 | 87.8 | — | -1.5 |
| | RTE | 6.5 | 89.9 | 89.2 | — | -0.7 |
| | WNLI | 0.0 | 54.9 | 54.9 | — | 0.0 |
| | SE14T1 | 50.6 | 86.9 | 90.5 | 84.8 | 3.5 |
| | MRPC | 32.0 | 67.9 | 68.7 | 66.9 | 0.8 |
| | DREAM | 0.0 | 81.1 | 81.4 | — | 0.3 |
| | Quartz | 24.5 | 79.2 | 77.7 | 83.9 | -1.5 |
| Factual Consistency Evaluation SummaC | CogenSumm | 2.0 | 88.4 | 88.6 | — | 0.2 |
| | XSumFaith | 0.7 | 74.6 | 75.2 | — | 0.5 |
| | PolyTope | 1.3 | 92.5 | 91.9 | — | -0.6 |
| | FactCC | 9.3 | 94.9 | 95.3 | — | 0.5 |
| | SummEval | 4.0 | 92.3 | 92.6 | — | 0.3 |
| | FRANK | 2.0 | 91.3 | 90.2 | — | -1.0 |
| Factual Consistency Evaluation TRUE | FRANK | 2.8 | 91.4 | 91.4 | — | 0.0 |
| | SummEval | 4.6 | 83.8 | 84.4 | — | 0.6 |
| | MNBM | 0.5 | 74.4 | 76.5 | — | 2.1 |
| | QAGS-C | 0.4 | 89.0 | 89.0 | — | 0.0 |
| | QAGS-X | 0.4 | 83.2 | 83.1 | — | -0.1 |
| | BEGIN | 27.9 | 81.1 | 81.0 | 82.3 | -0.1 |
| | Q2 | 41.6 | 79.2 | 81.3 | 77.8 | 2.1 |
| | DialFact | 28.9 | 85.1 | 85.2 | 89.0 | 0.1 |
| | PAWS* | 19.4 | 98.4 | 98.8 | 97.1 | 0.4 |
| | FEVER* | 54.4 | 94.9 | 96.2 | 94.9 | 1.3 |
| | VitaminC* | 17.5 | 98.3 | 98.3 | 99.6 | -0.1 |
| Factual Consistency Evaluation Other | XSumFaith | 0.8 | 28.8 | 30.1 | — | 1.3 |
| | SummEval | 3.3 | 66.7 | 64.9 | — | -1.8 |
| | Q-Xsum | 0.4 | 53.9 | 53.6 | — | -0.2 |
| | Q-CNNDM | 0.4 | 76.1 | 76.1 | — | 0.0 |
| | F-Xsum | 2.6 | 38.9 | 45.7 | — | 6.8 |
| | F-CNNDM | 2.6 | 68.9 | 67.9 | — | -1.0 |
| | SamSum | 0.0 | 47.7 | 47.7 | — | 0.0 |
| Average | | | | | | -0.1 |

