# OpenReview forum: "Text Alignment Is An Efficient Unified Model for Massive NLP Tasks"
_NeurIPS.cc/2023/Conference — NeurIPS 2023 poster_

### Official Review · Reviewer_Bk1X · 2023-07-05

**Soundness:** 4 excellent
**Presentation:** 4 excellent
**Contribution:** 4 excellent
**Rating:** 7
**Confidence:** 4

**Summary:**

While next word prediction produces task-general models that can do a wide variety of tasks when prompted, it is not an efficient formulation in that it requires very large models. On the other hand, fine-tuned models achieve higher performance at smaller sizes but are specific to a few tasks. This paper proposes text alignment as a middle ground that encompasses a wide range of tasks while allowing for smaller models than next word prediction. Concretely, they convert 28 datasets (encompassing tasks like entailment, IR, QA, coref, and consistency) into the text alignment format and fine-tune RoBERTa on them. The resulting model outperforms much larger models (that are instruction-finetuned), as well as RoBERTa with task-specific fine-tuning.

**Strengths:**

(1) The idea of using text alignment as a task-general interface is interesting and seems useful for producing useful task-general models at smaller sizes.

(2) The experiments are extremely thorough, and the method performs well across the board.

(3) The paper is well-written and clear.

**Weaknesses:**

While this section contains some suggested experiments, I support the acceptance of this paper regardless of whether or not they are run during the rebuttal period.

(1) While the paper frames alignment as being more general than multi-task finetuning, they only evaluate the model on tasks seen during alignment finetuning. Indeed, as shown in the task ablations in Table 5, it seems that the model can only do tasks that are included during training. Therefore, I wonder how different the model is from simply doing multi-task fine-tuning, and what clear advantages are provided by using a unified interface. While already strong, I think the results would be even stronger if there were examples of the model doing unseen tasks.

(2) Related to questions of how general alignment is, the tasks included in the training and evaluation feel very close to entailment, which might be a key factor in enabling them all to use the same interface without task-specific heads. Therefore, I wonder how performance would be affected if tasks very far from entailment were included. (For example, you could include POS tagging, where x_1 is the original sentence and x_2 is the sentence with some or all of the words replaced by their part of speech.)

(3) Related to the above, while the paper claims that changing the interface from next word prediction to alignment allows for smaller models, I wonder if the smaller model sizes are simply a result of considering a narrower set of tasks than the instruction-finetuned models. I suspect that FLAN-T5 needs larger model sizes simply because it needs more capacity to do more tasks. One relevant ablation testing this question would be to take the same RoBERTa model and instruction-finetune it on the same 28 datasets as the alignment model.

**Questions:**

(1) For handling longer contexts, I wonder if it would make sense to take min_j max_i f(x_1^i, x_2^j) instead of mean_j max_i f(x_1^i, x_2^j) as in the paper, with the interpretation that all of the facts in x_2 should be supported. The mean could fail if x_2 contains mostly supported statements, except for an egregiously unsupported statement in a single chunk.

**Limitations:**

Addressed

---

> ### Author Rebuttal · Authors · 2023-08-10
>
> Thank you for your comments. Your recognition of our ALIGN framework, experiments and paper writing is truly encouraging. We also appreciate your list of suggested experiments.
>
> **Performance on unseen tasks**
>
> Despite we group some training and evaluation datasets into the same type, they can have significantly different formats. For example, the DREAM dataset (used for evaluation) contains QA samples with dialogues contexts, while in our training datasets the QA contexts are usually articles (no dialogues). The TRUE benchmark (factual consistency evaluation, Figure 4) also contains dialogue datasets and we do not have similar (dialogue) datasets in our training data.
>
> As correctly pointed out by Bk1X, our ablation study indicates the inclusion of tasks (e.g., NLI, QA, etc.) during alignment training significantly boosts model performance on datasets of that task. However, as shown in the factual consistency evaluation use case (see Table 3 and Figure 4), our method outperforms existing approaches when good quality training data for the task (e.g., automated machine summarization evaluation) is limited.
>
> We also included an additional comparison between our method and multitask learning in the general response. The experiment results suggest that our unified interface allows the model to better generalize to new datasets in the zero-shot setting.
>
> **Alignment in relation to entailment**
>
> Indeed, we consider entailment to be a special case of the more general definition of alignment. We note we don’t claim that alignment can be used to solve all NLP problems. We intentionally choose a set of related tasks such that they can be effectively modeled and learned using a single interface. Our experiments show this is a good tradeoff. That said, we are excited to explore if alignment can be generalized to more distant tasks such as POS tagging.
>
> **smaller model, narrower tasks**
>
> We would like to thank the reviewer for bringing up the ablation experiment. However, in this experiment, we decide to instruction fine-tune a a T5 model due to the following reasons: 1) Roberta is an encoder-based transformer model, and it does not undergo training involving next word prediction,  2) existing instruct finetuning methods are designed for decoder-based models such as T5 and PALM, 3) this choice will enable us to effectively compare the distinctions between the two training objectives: next word prediction and alignment. We use the T5-base (250M parameters) version to show whether a smaller instruction fine-tuned model could still perform well on the alignment tasks, compared with the alignment-based model.
>
> We instruction-finetune the T5-base model on the same datasets as our alignment model. We don’t convert QA tasks since T5 naturally supports sequence generation. We follow the prompts mentioned in [1, 2] and format the datasets to feed the T5 model. The following tables are the result for the instruction finetuned T5-base.
>
> Table 1. Comparison with the instruction finetuned next word prediction model on in-domain tasks
> |Model\EvaluationTasks|NLI|FactVerification|STS|Paraphrase|QA|Coreference|Average|
> |-|--|-|--|-|--|-|--|
> |Alignment-base(125M)|**70.9**|**83.3**|**89.9**|**91.4**|**78.2**|**81.4**|**82.5**|
> |Instruction-finetuned-T5(222M)|53.7|68.7|62.0|83.2|30.3|42.9|56.8|
>
> Table 2. Comparison with the instruction finetuned next word prediction model on zero-shot tasks
> |Dataset\Model|Alignment-base|Inst-FT-T5-base|
> |-|-|-|
> |AXB|**75.1**|65.3|
> |AXG|**59.8**|50.8|
> |CB|**76.8**|58.9|
> |RTE|**83.4**|59.9|
> |WNLI|**52.1**|47.9|
> |SE14T1|**90.7**|48.5|
> |MRPC|66.0|**66.1**|
> |DREAM|**71.3**|35.8|
> |Quartz|**59.7**|50.6|
> |AVG|**70.5**|53.8|
>
> The result shows the alignment-based model achieves better performance than the instruction finetuned model on the in-domain and zero-shot tasks, showing the effectiveness of the alignment model when training on a set of alignment tasks. Therefore, the good performance does not come from narrowing the tasks as the small instruction finetuned T5-base performs worse than the alignment model.
>
> Reference
> [1] Chung, Hyung Won, et al. "Scaling instruction-finetuned language models." arXiv preprint arXiv:2210.11416 (2022).
> [2] Longpre, Shayne, et al. "The flan collection: Designing data and methods for effective instruction tuning." arXiv preprint arXiv:2301.13688 (2023).
>
> **Aggregation method (min-max vs. mean-max)**
>
> We agree that, if the model used for alignment estimation is perfect, using a min-max formulation as suggested by the reviewer would more accurately reflect the definition of alignment. However, in practice, the alignment estimator can be noisy, and we speculate taking the average helps remove some of that noise. In comparison, the min-max setup would be easily affected by underestimation. We also note mean-max aggregation is widely used in previous automatic evaluation work, such as SummaC [1] and SMART [2].
>
> We have therefore experimented with the min-max aggregation method on the factual consistency tasks. The results are shown as follows:
>
> ||Alignment-base|-base-minmax|Alignment-large|-large-minmax|
> |:-|:-:|:-:|:-:|:-:|
> |SummaC|85.9|86.1|89.0|89.1|
> |TRUE|**85.3**|84.2|**87.2**|86.2|
> |Other-Pearson|**45.3**|43.7|**54.4**|52.1|
> |Other-Spearman|42.0|**42.3**|**49.7**|49.2|
> |Other-Kendall|33.1|33.1|**39.9**|38.8|
>
> The bold number indicates a better performance among the same sized alignment models. It shows our mean-max aggregation outperforms min-max in most cases.
>
> Reference
> [1] Laban, Philippe, et al. "SummaC: Re-visiting NLI-based models for inconsistency detection in summarization." Transactions of the Association for Computational Linguistics 10 (2022): 163-177.
> [2] Amplayo, Reinald Kim, et al. "SMART: sentences as basic units for text evaluation." arXiv preprint arXiv:2208.01030 (2022).

---

> > ### Comment · Reviewer_Bk1X · 2023-08-15
> >
> > Thanks for the very thorough response. All of my questions were answered.

---

### Official Review · Reviewer_zhhb · 2023-07-05

**Soundness:** 3 good
**Presentation:** 4 excellent
**Contribution:** 3 good
**Rating:** 4
**Confidence:** 4

**Summary:**

This work proposed a text alignment model for a wide range of tasks that aims to measure the degree of alignment between their information. To be more specific, 5.9M examples from 28 datasets are used to fine-tune RoBERTa model. Experimental results show that the text alignment-enhanced model delivers comparable or superior performance compared to larger LMs, validating the effectiveness of the proposed method.

**Strengths:**

1. 5.9M examples from 28 datasets are extracted for LMs fine-tuning, and the experimental result on in-domain datasets (Table 1) and zero-shot setup (Table 2) demonstrate the effectiveness of Alignment-RoBERTa.
2. Experimental code has been submitted, benefitting future pre-fine-tuning research.

**Weaknesses:**

1. The previous pre-fine-tuning work proposed combining multiple losses with different weights [1], which diminishes the technical contribution of this work in terms of pre-fine-tuning.
2. Additional synthetic data is needed, increasing the complexity of the proposed pre-fine-tuning method. Also, the ablation study and corresponding are insufficient in terms of how much contribution synthetic data makes to the overall performance.
3. Experimental setup is not convincing because of the contamination of the training set (seen task) and test set (unseen task). Task clustering is needed to remove the concern.


[1] https://aclanthology.org/2021.emnlp-main.468.pdf

**Questions:**

As for Flan models, there is a potential concern regarding contamination of the training set and test set, especially when it comes to the in-domain test, as shown in Table 1 and Figure 2. I understand you have seen and unseen datasets. However, the training and test sets may not have the exact same data points, but they share similar patterns or characteristics. In this case, things become very trivial: the improvement is just because your 5.9M contains similar data instances as in the test set. Given this context, I have two questions:

1. Did you experiment with an out-of-domain test setting for the alignment-based RoBERTa model? I'd like to know if you cluster the tasks to determine the task boundary, as many tasks have a high overlap.
2. Line 634 in the appendix: you mentioned for each dataset, we only use the first 500k samples. How did you determine the number "500k," and how will performance change if you vary this number?

**Limitations:**

The authors state the limitation regarding biases in the constructed dataset, and it is an inevitable issue in the training data.

---

> ### Author Rebuttal · Authors · 2023-08-10
>
> **Previous Pre-finetuning Works**
>
> We’d like to clarify that our text alignment method is **not** pre-fine-tuning. Instead, we develop a unified alignment model that is directly applied to a diverse set of tasks **without any additional finetuning**. As we pointed out in the related works section, our work differs from MUPPET (Aghajanyan et al.) in that we 1) use a unified definition of alignment instead of multitask learning , 2) share all model components across tasks and do not use dataset-specific heads, and 3) our model can be directly applied to different tasks without additional finetuning.
>
> **Contribution of synthetic data**
>
> The method we use to create synthetic data is easy and commonly used by many pretrained models (Kryściński et al., 2019; Deng et al., 2021). We also note that some popular datasets we use already contain synthetic samples (e.g., PAWS, DocNLI).
>
> We acknowledge the need for further ablation study on the synthetic data and we thank the reviewer for the suggestion. The experiment results for removing the synthetic data are shown below.
>
> Table 1: Performance comparison on NLU tasks
> | Training \ Evaluation Tasks | NLI      | Fact Verification | STS      | Paraphrase | QA       | Coreference | Average  |
> | --------------------------- | -------- | ----------------- | -------- | ---------- | -------- | ----------- | -------- |
> | All tasks                   | **70.9** | **83.3**          | 89.9     | 91.4       | 78.2     | 81.4        | 82.5     |
> | -Synthetic                  | 70.4     | 83.1              | **90.1** | **92.0**   | **78.6** | **83.1**    | **82.9** |
>
>
> The result shows that the synthetic data is not critical as it does not make a big difference to the model performance. The synthetic samples are included to provide a more comprehensive representation of the data that the alignment model may encounter. Therefore, the synthetic datasets are not a fundamental part in our alignment framework and they will not significantly increase the complexity of it (the synthetic samples are minor in the whole training set), which in turn shows the efficiency and simplicity of our proposed method.
>
> **Test set contamination**
>
> Even for tasks of the same type (e.g., QA), over evaluation datasets can have very different patterns from the training datasets. For instance, the DREAM question answering dataset we use for zero-shot evaluation includes dialogues as the QA context, while training QA datasets contain general QA samples (e.g., SQuAD v2). In our factual consistency evaluation use case, the evaluation benchmarks also contain task types not present in training data. For example, TRUE contains dialogue datasets that have very different characteristics compared to NLI, QA, etc. data used for training. In both cases, our experiment results indicate that our model generalizes better than the baselines.
>
> **First 500k samples**
>
> We limit the number of samples per training dataset to avoid extreme data imbalance between datasets as some of them are extremely large (please refer to Table 8 in the appendix for statistics). Our practice follows previous work (e.g. FLAN-T5 (Chung et al., 2022)) that also uses similar methods to mitigate the dominance of large datasets. Additionally this limit helps us save computational resources. It’s certainly possible that including more training data could lead to better performance, and we are happy to explore more in the future
>
> Reference
> [1] Kryściński, Wojciech, et al. "Evaluating the factual consistency of abstractive text summarization." arXiv preprint arXiv:1910.12840 (2019).
> [2] Deng, Mingkai, et al. "Compression, transduction, and creation: A unified framework for evaluating natural language generation." arXiv preprint arXiv:2109.06379 (2021).
> [3] Chung, Hyung Won, et al. "Scaling instruction-finetuned language models." arXiv preprint arXiv:2210.11416 (2022). https://arxiv.org/pdf/2210.11416.pdf, page 46

---

> > ### Comment · Reviewer_zhhb · 2023-08-21
> >
> > Thank you for your answers. The term "Pre-fine-tuning" is used in the title of [1]. It herein refers to using 5.9M data instances (in your case) to further tune the model and test in the downstream tasks without additional task-specific fine-tuning. So I think I get this right.
> >
> > Your answer partially addresses my concerns. However, you have not addressed the concern over test set contamination. In the previous FLAN paper [2], section 2 stated that "we group datasets into clusters by task type and hold out each task
> > cluster for evaluation while instruction tuning on all remaining clusters." The reason for doing this is straightforward: training on one QA dataset, for example, might help the model do better on another QA dataset. Therefore, they group all datasets into clusters by type of task and hold out not just the single QA dataset but the entire QA task cluster to which the dataset belongs.
> >
> > In contrast, in your case, the QA datasets appear in both your training data (Table 1) and test data (Table 2). That's why I have this concern regarding training/test set contamination. The model may learn very simple superficial cues/patterns in the training set without literally understanding them. In short, seen/unseen datasets SHOULD be determined by task cluster instead of a single task.
> >
> > [1] https://aclanthology.org/2021.emnlp-main.468.pdf
> > [2] https://arxiv.org/pdf/2109.01652.pdf

---

> > > ### Author Response · Authors · 2023-08-21
> > >
> > > Thanks for your feedback.
> > >
> > > **Pre-fine-tuning**
> > >
> > > We’d like to clarify that the term “pre-fine-tuning” as used in [1] does not refer to what you described. In particular, **“pre-fine-tuning” DOES require additional task-specific fine-tuning** before applying the “pre-fine-tuned” model to a downstream task. In contrast, as you pointed out, **our method** applies to downstream tasks **WITHOUT** additional task-specific fine-tuning. Our work is thus fundamentally different from “pre-fine-tuning” methods as in [1].
> > >
> > > More specifically, the work [1] explicitly uses task-specific finetuning after “pre-fine-tuning”:
> > >
> > > 1) The MUPPET authors clearly state in the abstract they *“propose pre-finetuning, an additional large scale learning stage **between** language model **pre-training** and **fine-tuning**”*. Furthermore, in the experimental setup, they mention that *“We first show that pre-finetuning improves the representations of pre-training models. To do so, we **fine-tune** our pre-finetuned models on a large set of tasks”* and *“**Finetuning** Outside of Pre-Finetuning Domain”*. These statements highlight the need for finetuning in the pre-finetuning method, which our alignment model does not require.
> > > 2) In 3.5 Experimental Setup of the MUPPET paper, the authors mention that *“Every Sentence Prediction dataset gets a **separate** classification head, for Commonsense and MRC we utilize a separate unified head for each task.”* However, our alignment model does not necessitate assigning a specific head for each downstream task.
> > >
> > >
> > > **Test Set Contamination**
> > >
> > > **There could be different ways of “grouping” and distinguishing between seen and unseen tasks/datasets.** For example, the FLAN paper groups tasks/datasets based on the *task type* as you mentioned. Under this perspective, the task of *factual consistency evaluation* (i.e., predicting a factual consistency score, Section 4.2) is of a different type than any training tasks (e.g., QA), and our model shows substantial improvement over diverse baselines on as many as 23 datasets. On the other hand, as we mentioned in the initial response, one could also define unseen or out-of-domain tasks/datasets as those of different textual patterns/characteristics than training tasks/datasets. For example (as mentioned in the initial response), the dialogue datasets in *factual consistency evaluation* (Section 4.2) are of very different patterns/characteristics than training data (and they are also of different task types); the DREAM QA dataset in Section 4.1 also involves very different forms of context (i.e., dialogue) compared to the QA tasks in training. **In either way, our rich experimental results have shown that our approach does generalize to unseen tasks and show strong improvements over diverse baselines.** Reviewer Bk1X has commended our rich experiments, while both reviewer ayzf and 6RaC concur on our strong results. We will make this clearer.
> > >
> > >
> > >
> > > **Reference**
> > >
> > > [1] Aghajanyan, Armen, et al. "Muppet: Massive Multi-task Representations with Pre-Finetuning." Proceedings of the 2021 Conference on Empirical Methods in Natural Language Processing. 2021.

---

> > > > ### Comment · Reviewer_zhhb · 2023-08-21
> > > >
> > > > Thank you for your answers.
> > > > Regarding the term "pre-fine-tuning," I was referring to your case rather than [1].
> > > > For test set contamination, you mentioned, "one could also define unseen or out-of-domain tasks/datasets as those of different textual patterns/characteristics than training tasks/datasets..." Then, it would be necessary to justify why you think your textual patterns/characteristics-based method is better than the task category-based method (used in [2]) regarding seen/unseen split. Alternatively, it is better to have a quantitative analysis of your division between seen and unseen tasks to endorse your strong performance improvement, such as high-order n-gram overlap checking in [3] (see section 4) or manual checking in the PaLM model [4] (see section 8).
> > > >
> > > > Therefore, I stick to my previous score.
> > > >
> > > > [1] https://aclanthology.org/2021.emnlp-main.468.pdf
> > > > [2] https://arxiv.org/pdf/2109.01652.pdf
> > > > [3] Language Models are Few-Shot Learners
> > > > [4] PaLM: Scaling Language Modeling with Pathways

---

> > > > > ### Author Response · Authors · 2023-08-21
> > > > >
> > > > > Thank you for your feedback again.
> > > > >
> > > > > **(1) "pre-fine-tuning"**
> > > > >
> > > > > It looks like we've already addressed your original concerns regarding the novelty of our work (i.e., the weakness-1 in your original comment). It seems we both agree our work is different from [1] as we don't require additional task-specific finetuning. Please let us know if you have any further questions in this regard.
> > > > >
> > > > > **(2) contamination**
> > > > >
> > > > > We'd like to clarify again that we have evaluated on "unseen" tasks/datasets of both **unseen task types** and **unseen patterns/characteristics**. We've given concrete examples of **both** cases in both our initial response and our second response. We were not arguing one way of seen/unseen splitting is better than the other -- We have evaluated and shown strong improvements in **both** ways.

---

### Official Review · Reviewer_6RaC · 2023-07-06

**Soundness:** 3 good
**Presentation:** 3 good
**Contribution:** 3 good
**Rating:** 7
**Confidence:** 4

**Summary:**

This paper leverages the fact that a lot of popular comparison based NLP tasks like entailment, paraphrase detection, semantic similarity judgement, multiple choice passage based QA etc. amount to learning a specific similarity function between two sets of sequences that is a proxy for “information alignment” between the two sets. Hence, this paper gathers together public datasets for many such tasks and with some light rule-based data augmentation results in 5.9 training examples from 28 datasets. Then a moderately sized RoBERTa model is finetuned on this big comparison-based dataset which is then compared to task specific models and larger models like FLAN T5 on many such datasets. This model has also been used as a metric for measuring factual consistency of NLG models like summarization models. The authors also use this model to detect questions that are unanswerable from the accompanying context to boost performance of systems on some QA datasets.


**Strengths:**

– The paper is well-motivated and the large aggregated dataset and the model trained on it will be useful to the community for further study.

– This model outperforms larger general models like FLAN T5 and is competitive with task specific finetuned RoBERTa models on various semantic comparison tasks (some of them unseen during training) which shows the effectiveness of similarity between various such tasks.

– The results on summarization evaluation are promising and the usage of this model for identifying unanswerable questions is interesting.

– The ablation study shows an interesting trend indicating that various comparison-based datasets and tasks are very similar and compatible with each other. This raises interesting questions related to the nature of these tasks and datasets.


**Weaknesses:**

– No comparison is made against specialized task-specific models. While fine-tuning RoBERTa on task-specific datasets is informative, a deeper insight into how the proposed model fares in comparison to more focused task-specific models will strengthen the comparison.

– While this paper focuses on comparison and understanding tasks, it is compared to larger generative models that are specially designed for natural language generation. While these models show impressive performance on these understanding/comparison tasks, a more informative comparison would be against larger encoder-based models that are specifically trained to do well on these datasets and benchmarks like SuperGLUE.


**Questions:**

See above

**Limitations:**

See above

---

> ### Author Rebuttal · Authors · 2023-08-10
>
> **Comparison with task-specific models**
>
> The goal of our work is to design a model that can perform well on a range of tasks without further task-specific fine-tuning. As a result, we compare with LLM that can be used in a similar way (do not require fine-tuning) and include the task-specific fine-tuned RoBERTa baseline as a sanity check. To the best of our knowledge, the results reported on leaderboards such as SuperGLUE are for models specifically fine-tuned on each of the datasets in the benchmark, making them not suitable for a direct comparison with our method.
>
> **Comparison with large encoder-based models**
>
> There are few large encoder-only language models, and we compare with the most well known large encoder-based model Megatron BERT using results reported by Shoeybi et al. (2020). Our alignment model has similar performance compared to Megatron BERT of a similar size, but the alignment model does **not** require further fine-tuning, while Megatron BERT does. We also note as the parameter count of Megatron BERT increases, its performance improvement is limited. This diminishing return is also one of the reasons that we opt to use a smaller model when it comes to the generality-vs-efficiency tradeoff.
>
> |         | Megatron | Megatron | Megatron | Alignment base | Alignment large |
> |---------|----------|----------|----------|----------------|-----------------|
> | size    | 336M     | 1.3B     | 3.9B     | 125M           | 355M            |
> | MNLI-m  | 89.7     | 90.9     | 91.4     | 87.82          | 90.34           |
> | MNLI-mm | 90.0     | 91.0     | 91.4     | 87.54          | 90.31           |
> | QQP     | 92.3     | 92.6     | 92.7     | 90.07          | 91.27           |

---

> > ### Comment · Reviewer_6RaC · 2023-08-16
> > **thanks for the rebuttal**
> >
> > The author response was informative and i am keeping my current score.

---

### Official Review · Reviewer_ayzf · 2023-07-07

**Soundness:** 3 good
**Presentation:** 3 good
**Contribution:** 2 fair
**Rating:** 5
**Confidence:** 4

**Summary:**

This paper proposes a way to cast a variety of classification tasks into a single text alignment task. The authors found out that using the text alignment task could generate better results on certain downstream tasks, compared to Flan-T5 and GPT-3.5.

**Strengths:**

The paper presents a novel approach by framing everything as an alignment task. The results are good, and I think the verifier results are interesting.

**Weaknesses:**

I understand that the text-alignment model is not well-suited for generative tasks. Therefore, I believe that the conclusion of the paper is a bit unfair to Flan T5 and Gpt-3, which can be used for generative tasks. It is well-known that sometimes specialized small models can be performed better than by much larger general models. Somehow, the proposed method sounds like another trade-off. I think the proposed method is reasonable, but I believe that further research into different trade-offs would be beneficial.



**Questions:**

Also, more baseline should be compared to. If the model just train the the same subset of the task without casting them into alignment model, will the model still perform well?

**Limitations:**

The proposed model cannot handle generative tasks.

---

> ### Author Rebuttal · Authors · 2023-08-10
>
> **More baseline - multitask learning**
>
> We add comparison with multitask learning as suggested, where the model just trains with exactly the same set of tasks without casting into alignment format. Please see the General Response for results. The results show our alignment model has better performance than the conventional multi-task learning, along with other advantages.
>
> **Trade-offs: small vs large & general, discriminative vs generative**
>
> Our decision to use a small discriminative model is intentional, and we have explicitly discussed this generality-vs-efficiency tradeoff in the introduction section. One of our contributions is to propose a concrete approach in the spectrum of the generality-vs-efficiency tradeoff. We have shown through experiments that this trade-off is desirable as it achieves better performance on a significant set of tasks, with smaller parameter counts than other more general models, leading to a model suitable for many interesting applications (its usefulness is also acknowledged by reviewers Bk1x and 6Rac). The ability to handle generative tasks (while interesting in its own right) is outside of the scope of our definition of alignment.

---

### Author Rebuttal · Authors · 2023-08-10

We thank all reviewers for your thoughtful and positive comments. We're encouraged by your appreciation that our text alignment framework is well-motivated (6RaC), novel (ayzf), and interesting (Bk1X); the resulting model has strong performance (ayzf, 6RaC, zhhb), enables interesting use cases (6RaC), and is useful to the community (6RaC, zhhb, Bk1X); the experiments are extremely thorough (Bk1X), and reveal insights about the nature of comparison-based tasks (6RaC); and lastly, the paper is well-written and easy to follow (Bk1X).

&nbsp;

**Here we address the concern of comparison with multitask learning**:

### Comparison with multitask learning
We add comparison with multitask learning and show the advantage of our unified alignment framework (we thank the reviewers for the suggestion). Specifically, to obtain the multitask-learning model, we follow the popular multi-task learning work (Muppet, Aghajanyan et al., 2021), and train the same base model with the same set of tasks/datasets as for our alignment model. Notably, different from our alignment model that uses a unified interface to accommodate all diverse tasks, the conventional multitask-learning model learns separate prediction heads for different tasks.

Results show our alignment model outperforms the multitask learning model on zero-shot datasets, while having similar performance on in-domain datasets. This suggests that our definition of alignment indeed helps the model better generalize to out-of-domain problems.

**Experiment results**

**(1)** The table below shows the **results for in-domain datasets**. The alignment model and the multitask learning model have roughly the same performance.
| Dataset | Multitask base | Alignment base |
|---|---:|---:|
| MNLI-mm | 87.61 | 87.54 |
| MNLI-m | 87.68 | 87.82 |
| ANLI-1 | 65.10 | 65.30 |
| ANLI-2 | 48.20 | 48.70 |
| ANLI-3 | 46.17 | 45.50 |
| SNLI | 91.33 | 90.78 |
| NLI-FEVER | 76.50 | 76.78 |
| VitaminC | 89.82 | 89.79 |
| SICK | 89.01 | 90.71 |
| STSB | 87.86 | 89.03 |
| PAWS | 93.94 | 92.33 |
| PAWS-QQP | 92.32 | 91.88 |
| QQP | 90.66 | 90.07 |
| Multi-RC | 83.83 | 82.20 |
| BoolQ | 81.32 | 81.07 |
| SciQ | 92.10 | 92.40 |
| GAP | 81.65 | 81.35 |
| **Average** | **81.48** | **81.37** |

**(2)** The table below shows the **results in the zero-shot setting**. For the multitask-learning model and each zero-shot evaluation task, we use a set of “reasonable” heads obtained during training for prediction, and we report the best, worst, and average performance among these heads. For NLI, QA, and paraphrase evaluation datasets, the “reasonable” heads are all heads trained with NLI, QA, and paraphrase detection datasets, respectively.

As shown in the table below, the alignment model outperforms the average performance of the “reasonable” multitask learning prediction heads, and the multitask learning model only slightly outperforms the alignment model if we **cherry-pick** the best performing head for each zero-shot dataset. In a strict zero-shot setting, as we can’t use evaluation data for head selection, and the best performing head varies across evaluation datasets, the reported “best head” performance is an unrealistic upper bound. Thus the results indicate the alignment model has an advantage over multitask learning when applied to new datasets.

| Dataset | Multitask base (avg. of heads) | Multitask base (best head) | Multitask base (worst head) | Alignment base |
|---|:---:|:---:|:---:|:---:|
| AXB | 74.94 | 76.18 | 72.37 | 75.09 |
| AXG | 63.58 | 65.73 | 60.39 | 59.83 |
| CB | 80.00 | 83.93 | 76.79 | 76.79 |
| RTE | 81.41 | 81.95 | 81.23 | 83.39 |
| WNLI | 56.34 | 60.56 | 47.89 | 52.11 |
| SE14T1 | 57.86 | 58.45 | 57.17 | 90.72 |
| MRPC | 69.60 | 71.19 | 68.23 | 65.97 |
| DREAM | 67.04 | 73.98 | 57.03 | 71.34 |
| Quartz | 59.09 | 63.52 | 54.46 | 59.69 |
| **Average** | **67.76** | **70.61** | **63.95** | **70.55** |



**In addition** to the above stronger performance in the zero-shot setting, there are two notable advantages of our proposed method over multitask learning:
1. Our approach casts all the diverse tasks into a unified alignment format, which allows us to leverage these datasets naturally. In comparison, it’s not straightforward for a discriminative multitask-learning model to train on some of the datasets in our alignment data. For example, a multitask-learning model cannot easily train on open-ended QA datasets where the ground truth answers do not exist in the context. For fair comparison, we reuse our converted alignment version of those datasets in multitask learning.
2. As the multitask-learning model uses task-specific heads, it cannot be straightforwardly applied to new out-of-domain datasets. In the out-of-domain (zero-shot) setting, for each of the new dataset we have to find the most “similar” training task and use the associated head for inference, which can be difficult when data is limited. In comparison, our alignment model has unified  prediction heads across tasks and datasets, making it easy to use the model for new datasets.

We’ll include the above results and articulate all experimental details in the revised version of the paper.

---

### Decision · Program_Chairs · 2023-09-21

**Decision:**

Accept (poster)

**Comment:**

Meta Review:

This paper proposes a text alignment model for a wide range of tasks that aims to measure the degree of alignment between their information. The authors leverage 5.9M examples from 28 datasets to fine-tune a RoBERTa model, which delivers comparable or superior performance compared to larger language models.

Strengths:
- The novel approach of framing everything as an alignment task shows promising results.
- The model outperforms larger general models like FLAN T5 and is competitive with task-specific fine-tuned RoBERTa models on various semantic comparison tasks.

Weaknesses:
- The proposed model cannot handle generative tasks, and the conclusion may be a bit unfair to Flan T5 and GPT-3, which can be used for generative tasks.
- No comparison is made against specialized task-specific models, which would strengthen the comparison.
- The experimental setup may have contamination between the training set and test set, raising concerns about the generalizability of the model.

Overall, while there remain some concerns of this work, the strengths outweigh the limitations. But I hope in the revised version, the author could do further research into different trade-offs, comparisons with specialized task-specific models. Also, I hope the author can address the concerns about contamination between the training set and test set (something like a 6-gram overlap detection will make the results much more convincing).